# COSMOS: ARE PERFORMANCE–COST TRADEOFFS PREDICTABLE IN MODEL–STRATEGY SELECTION?

## ABSTRACT

Large language models (LLMs) achieve excellent performance across numerous tasks by using a diverse array of adaptation strategies. However, selecting the optimal combination of model, strategy, and configuration under resource constraints is challenging and typically requires extensive experimentation. We ask whether it is possible to accurately predict both downstream performance and cost without running expensive adaptation trials. We introduce COSMOS, a general analysis framework for approaching this *strategy selection problem* through low-cost prediction of adaptation outcomes. We instantiate our framework via a pair of powerful predictors: embedding-augmented lightweight proxy models to predict fine-tuning performance, and low-sample scaling laws to forecast retrieval-augmented in-context learning. Evaluations across eight representative benchmarks demonstrate that ***COSMOS instantiations achieve high prediction accuracy while reducing computational costs by 92.72% on average, and up to 98.71% in resource-intensive scenarios***. Our results show that efficient prediction of adaptation outcomes is possible, enabling practitioners to navigate large model–strategy–configuration spaces efficiently and substantially reduce deployment cost while maintaining strong task performance.

## 1 INTRODUCTION

Large language models (LLMs) have scaled dramatically in both capability and availability, with millions of models now shared on Hugging Face. Each model offers distinct performance characteristics and computational demands. The emergence of diverse adaptation techniques has further expanded the space of possible deployment configurations. This raises a *key challenge*: how can we ***systematically identify an optimal choice of model and adaptation strategy in a cost-effective way***?

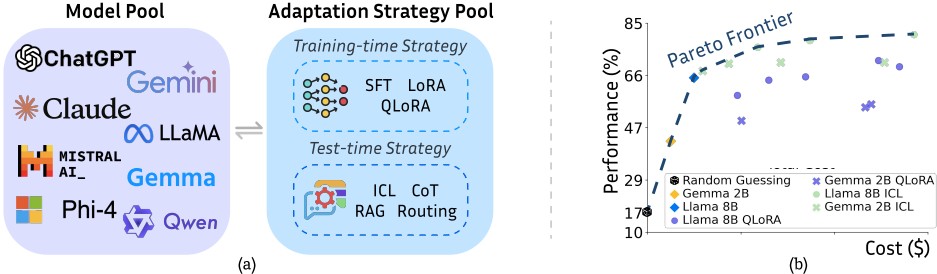

Figure 1: Overview of the strategy selection problem for LLMs and performance–cost tradeoff. **(a)** Given a downstream task, practitioners select from a pool of foundation models and adaptation strategies. **(b)** Each model–strategy combination results in different performance and cost. The challenge lies in choosing optimal combinations that balance performance and cost.

We formalize this as the strategy selection problem for LLMs: given a downstream task, identify the model–strategy combination that best balances performance and cost. Figure 1 illustrates this setting, where practitioners can select from a pool of foundation models and adaptation strategies, each with their own configuration space, to solve tasks across various domains. The challenge lies in navigating

this vast search space to find the best model–strategy combination without exhaustively evaluating all possibilities.

Intuitively, one way to tackle this problem is to *predict* how much each adaptation buys us. This motivates COSt-effective MOdel–Strategy prediction (COSMOS), a **unified analysis framework for approaching the strategy selection problem via predictive modeling**. COSMOS defines components of adaptation outcomes, and how performance and cost predictions can jointly inform strategy choice, obviating the need for running resource-intensive experiments.

This leads to two central research questions: **RQ1:** *are there sufficiently accurate predictors available to effectively use COSMOS in popular LLM adaptation scenarios?* **RQ2:** *are there benefits to custom predictors that exploit particular properties of adaptation approaches*—as opposed to general-purpose predictors that apply to all adaptation strategies?

To study these questions, we instantiate COSMOS on two prominent adaptation scenarios: 1) QLoRA fine-tuning, where we predict adaptation gains with an embedding-augmented lightweight proxy model, and 2) retrieval-augmented in-context learning, where we leverage observed scaling laws. The predictors for both scenarios *achieve excellent predictive performance*. We also show that while scaling law approaches to prediction can be used broadly, including for predicting fine-tuning performance, *specialized predictors can have substantially higher efficiency*.

Through extensive experiments across eight representative benchmarks spanning both general and specialized tasks, we demonstrate that instantiations of COSMOS achieves excellent prediction accuracy (mean absolute error of 1.09%) while reducing computational costs by an average of 92.72% (up to 98.71%) compared to exhaustive experimentation. This means that efficient prediction of adaptation outcomes is not only feasible, but can empower practitioners to navigate the large space of model–strategy combinations and make informed decisions that optimize both performance and cost.

Our main contributions are summarized as follows:

- We formalize and provide the **first** systematic study of the *strategy selection problem* for LLMs in a multi-model, multi-strategy (training-time and test-time), multi-configuration, multi-task, and cost-sensitive setting via predictive modeling.
- We introduce COSMOS, a general analysis framework that predicts the outcomes of adaptation strategies, enabling accurate and data-efficient estimation of both performance and cost across training-time and test-time strategies.
- Through extensive evaluation on eight diverse benchmarks (over 10,000 GPU hours, 7,700 experiments), we show COSMOS instantiations achieve superior prediction accuracy (1.09% MAE) while reducing computational cost by up to 98.71% compared to baselines.
- We obtain a number of new insights of independent value to practitioners, including the relative benefits of test-time vs. train-time adaptation strategies and the tradeoffs between universal and tailored performance predictors.

## 2 RELATED WORKS

**Adaptation strategies for LLMs.** Strategies for adapting pre-trained LLMs for downstream tasks broadly fall into two categories: training-time and test-time adaptation. Training-time strategies include supervised fine-tuning (Raffel et al., 2020; Wei et al., 2021) and parameter-efficient variants such as LoRA (Hu et al., 2021) and QLoRA (Dettmers et al., 2024). Test-time methods such as in-context learning (Brown et al., 2020), advanced search and prompting (Wei et al., 2022; Yao et al., 2023), and decoding (Park et al., 2024) are popular (Welleck et al., 2024). While these individual strategies are effective, efficiently selecting and configuring them jointly is underexplored.

**Model routing.** Routing sends easier queries to cheaper models and reserves powerful models for harder queries (Chen et al., 2024; 2023; Šakota et al., 2024; Shnitzer et al., 2023; Yue et al., 2023). While routers can balance performance and cost in model selection, they usually operate with a fixed model pool and do not consider a broad space of adaptation strategies. In contrast, our work *goes beyond pure model selection and routing*.

**Scaling laws and performance prediction.** Training-time scaling laws (Hoffmann et al., 2022; Kaplan et al., 2020) explore the relationship between model size, training compute, and training/test loss. Recent work (Haowei et al., 2024; Zeng et al., 2025; Haowei et al., 2025) further use variants of

rectified scaling laws to model how SFT test loss changes with data size. He et al. (2025) similarly models data size vs. VLM alignment test loss. Recent work on test-time scaling Snell et al. (2024) has systematically analyzed how performance gains from inference strategies such as best-of-N sampling (Cobbe et al., 2021; Lightman et al., 2023) and beam-search (Feng et al., 2023; Yao et al., 2023) scale with task difficulty. Similarly, Ruan et al. (2024) identified systematic links between LLM performance and low-level skills. However, existing approaches are limited to coarse-grained, task-agnostic predictions that frequently fail to consider the interplay between training-time and test-time strategy adaptations. Another line of work trains predictors directly on historical multilingual translation results to forecast performance in new languages (Xia et al., 2020; Anugraha et al., 2025). Yet, these methods are task-specific, typically require large volumes of full historical observations or extensive cross-validation, support only narrow configuration spaces, and are not cost-aware. By contrast, our approach is *general-purpose, cost-aware, requires neither exhaustive historical data nor cross-validation*, and *supports diverse configuration inputs*. It is applicable across diverse domains, while also accounting for the full spectrum of costs—including prediction, adaptation, and evaluation. In this way, our methods deliver accurate predictions for both *performance and overall cost across multiple dimensions, therefore enabling efficient and cost-aware model–strategy selection.* A detailed comparison between COSMOS and recent scaling-law–based approaches is provided in Appendix R.

**AutoML, hyperparameter optimization, and NAS.** Model selection, training hyperparameter optimization (HPO), and the development of new architectures optimized for each task are the traditional domains of AutoML and Neural Architecture Search (NAS). Recent work focuses on lifting these approaches to modern LLM settings (Roberts et al., 2024; Saad-Falcon et al., 2024). Most of the techniques within these areas are either narrow (*e.g.*, HPO (Yu & Zhu, 2020) focuses on one specific type of adaptation) or extremely expensive (Roberts et al., 2021; Shen et al., 2023). *The approach we propose is simultaneously general and cost-efficient.* Our framework and techniques are compatible with tools from AutoML and NAS.

## 3 THE STRATEGY SELECTION PROBLEM AND COSMOS PREDICTION FRAMEWORK

We formalize the strategy selection problem (Section 3.1) and introduce COSMOS, a unified predictive framework for estimating performance and cost across adaptation strategies (Section 3.2). Finally, we describe a cost analysis methodology that enables fair, end-to-end comparison across diverse adaptation approaches (Section 3.3).

### 3.1 THE STRATEGY SELECTION PROBLEM

Our goal is to systematically identify good choices of model and adaptation strategy in a cost-effective way. To formalize this problem, we define a *strategy navigator* ($M_D$) whose purpose is to determine the optimal combination of model, adaptation strategy, and configuration for a downstream task $D \in \mathcal{D}$ that balances performance and cost-efficiency. Formally, we are given the following:

- A *model pool* $\mathcal{F} = \{f_1, f_2, \ldots, f_K\}$, where each model $f_k \in \mathcal{F} : \mathcal{X} \to \mathcal{Y}$ maps an input query $\mathbf{x} \in \mathcal{X}$ to a model answer $\hat{\mathbf{y}} \in \mathcal{Y}$,
- An *adaptation strategy pool* $\mathcal{T} = \{T_1, T_2, \ldots, T_J\}$,
- A configuration space $\Omega$ where $\omega \in \Omega$ specifies parameters for applying a strategy,
- A performance metric $\pi$ and cost function $c$.

Let $T_j^{\omega}(f_k)$ represent a model after applying strategy $T_j$ with configuration $\omega$, resulting in a performance-cost pair $(\pi(T_j^{\omega}(f_k)), c(T_j^{\omega}(f_k)))$. The navigator selects the optimal combination by solving:

$$M_D(\mathcal{F}, \mathcal{T}, \Omega) = \underset{f_k \in \mathcal{F}, T_j \in \mathcal{T}, \omega \in \Omega}{\arg\max} s(\pi(T_j^{\omega}(f_k)), c(T_j^{\omega}(f_k))), \quad (1)$$

where $s : \mathbb{R} \times \mathbb{R}_+ \to \mathbb{R}$ is a score function that captures the trade-off between performance and cost.

*Example* 1. Consider a sentiment analysis task where we have one 7B parameter model $f_1$ and two adaptation strategies: $T_{\text{probe}}$ (linear probing) and $T_{\text{tune}}$ (full fine-tuning). Each strategy's configuration space includes learning rate, number of epochs, and training data size, each with 10 possible values. This results in 2000 total combinations (1 model × 2 strategy x 1000 configurations per strategy).

**Computational cost of strategy selection problem.** One approach is to exhaustively try all combinations. However, this quickly becomes prohibitively expensive. This naturally raises the question: *can we solve the strategy selection problem cost-efficiently?* We answer this affirmatively by introducing COSMOS, a framework to solve the strategy selection problem via predicting adaptation gains, resulting in a cheaper way to explore the search space.

## 3.2 COSMOS: Adaptation Outcomes Prediction Framework

Given the computational challenges of the strategy selection problem, one intuitive approach is to employ a cheap predictor with a small amount of data rather than conducting expensive full-scale experiments. We formalize this predictive approach through COSMOS (COSt-effective MOdel–Strategy prediction).

For an adaptation strategy $T_j \in \mathcal{T}$ applied to a model $f_k \in \mathcal{F}$ with configuration $\omega \in \Omega$, COSMOS defines: 1) a performance predictor: $\hat{\pi}(T_j^\omega(f_k)) = P_{j,k}(\omega) \approx \pi(T_j^\omega(f_k))$, and 2) a cost predictor: $\hat{c}(T_j^\omega(f_k)) = C_{j,k}(\omega) \approx c(T_j^\omega(f_k))$. Here, $P_{j,k}$ and $C_{j,k}$ are strategy-specific predictors that map configurations to expected performance and cost respectively. These predictors can take various forms depending on the nature of the adaptation strategy $T_j$. Examples of predictors include a lightweight proxy model and calibration on a small validation set to estimate adaptation outcomes, *e.g.*, a lightweight proxy model with a small calibration step for fine-tuning approaches (*e.g.*, fine-tuning); a scaling-law–based extrapolator using early measurements for test-time strategies.

Ideal performance and cost predictors should satisfy two properties: 1) *Cost-efficiency*: $c_{\text{predict}}(P_{j,k}, C_{j,k}) \ll c_{\text{adapt}}(T_j^\omega, f_k)$. The prediction cost must be significantly lower than the actual adaptation. And 2) *Strategy-awareness*: $P_{j,k} \in \mathcal{P}_j, C_{j,k} \in \mathcal{C}_j$, where $\mathcal{P}_j$ and $\mathcal{C}_j$ are the sets of valid performance and cost predictors, respectively, for the strategy $T_j$. While not strictly required, predictors that exploit strategy-specific signals often yield superior prediction accuracy and cost efficiency in practice.

The resulting predicted outcomes feed directly into the scoring function used to select the best model–strategy–configuration.

**Cost analysis.** Costs are: 1) Prediction cost: $c_{\text{predict}}(P_{j,k}, C_{j,k})$, including strategy-specific prediction overhead and calibration cost using validation data if necessary; and 2) Selected strategy cost: $c(T_{\hat{j}}^{\hat{\omega}}, f_{\hat{k}})$, including the cost of applying the chosen strategy and final evaluation cost, detailed in Sec. 3.3. The framework is efficient when $\sum_{j,k} c_{\text{predict}}(P_{j,k}, C_{j,k}) + c(T_{\hat{j}}^{\hat{\omega}}, f_{\hat{k}}) \ll \sum_{j,k,\omega} c(T_j^\omega, f_k)$.

**Framework instantiation.** To apply this framework to a specific adaptation strategy $T_j$, one needs to: 1) Choose an appropriate predictor type based on strategy characteristics; 2) Design the predictor architecture or model; 3) Define the prediction cost calculation. In Section 4, we do so for two diverse adaptation strategies: 1) QLoRA fine-tuning: predict performance via an embedding-augmented linear proxy model; and 2) retrieval-augmented ICL: predict performance via observed scaling law. We consider the full spectrum of cost based on both computing-based and token-based methods.

*Example* 2. Continuing with the sentiment analysis task from Ex. 1, For $T_{\text{tune}}$ (full fine-tuning), the predictor $P_{\text{tune}, f_1}$ could be a lightweight linear model trained on frozen embeddings and calibrate the performance from a small validation set. Assume the total cost of prediction including the training cost of the proxy model and validation cost arising from performance calibration, for total 1000 configs prediction, the prediction costs \$5, however, the actual total cost of adaption can be \$500. This demonstrates the efficiency property as $c_{\text{predict}}$ (\$5) $\ll c_{\text{adapt}}$ (\$500).

**Adaptation Strategies.** In Appendix Q, we list a variety of adaptation strategies, including training-time, test-time, and hybrid adaptation strategies. We include model routing as a special case and study it experimentally in Section 5.

## 3.3 Cost Analysis Framework

The effectiveness of an adaptation strategy must be evaluated jointly with its computational cost. To support practical, cost-aware strategy selection, we consider and model all costs incurred during the adaptation, evaluation, and prediction phases. Costs may be expressed in FLOPs, wall-clock time, energy, monetary units, or any consistent internal measure.

**Total cost.** Given a specific task $D$, the total cost for any adaptation strategy $T_j^\omega$ configured by $\omega$, applied to model $f_k$ comprises two components:

$$c(T_j^\omega, f_k) = c_{\text{adapt}}(T_j^\omega, f_k) + c_{\text{eval}}(T_j^\omega(f_k), D), \tag{2}$$

where $c_{\text{adapt}}$ represents the adaptation cost (*i.e.*, cost of applying the adaptation strategy) and $c_{\text{eval}}$ (*i.e.*, cost of evaluating adapted model performance).

**Strategy-specific adaptation cost.** Different strategies incur adaptation costs $c_{\text{adapt}}$ through different mechanisms: test-time strategies incur inference-level costs, which may be measured using token usage, FLOPs, wall-clock time, or other inference-related metrics, and scale with the number of inference passes. Training-time strategies incur training costs, which can be computed using: 1) computing-based method: (e.g., GPU/TPU hours, FLOPs, energy, wall-clock time); 2) token-based metrics (e.g., training tokens × epochs); or 3) any domain-specific cost model appropriate for the deployment environment.

**Prediction cost.** Prediction cost covers the computation needed to produce performance and cost estimates via predictors $P_{j,k}$ and $C_{j,k}$. This may include: 1) lightweight proxy model training for performance prediction, 2) calibrating performance on validation data if necessary, and 3) strategy-specific overheads, such as obtaining a few early observations for scaling-law fitting. Cost prediction typically leverages the same validation runs or dataset information used for performance prediction, thus incurring minimal additional overhead: $c_{\text{predict}}(P_{j,k}, C_{j,k}) = c_{\text{proxy}} + c_{\text{overhead}}(T_j^\omega, f_k) + c_{\text{val}}(D_{\text{val}})$. This unified formulation enables consistent comparison across heterogeneous adaptation approaches and supports the cost-aware strategy selection enabled by COSMOS (Section 3.2).

# 4 STUDYING COSMOS: POPULAR SCENARIOS AND GENERAL VS. SPECIFIC PREDICTORS

We use COSMOS to study a pair of important questions: ***First***, do there exist efficient instantiations of COSMOS for prominent LLM adaptation scenarios? ***Second***, should we rely on a universal predictive approach that can be applied broadly across adaptation strategies—or can tailored prediction methods produce superior results when available? We study these via the following scenarios.

**Instantiation setup.** We explore two popular complementary adaptation strategies, each paired with a prediction approach—one tailored and one general. The first strategy is QLoRA fine-tuning $T_{\text{QLoRA}}^{\text{tr}}$; we predict performance using an embedding-augmented linear model. The second is retrieval-based in-context learning (ICL) $T_{\text{ICL}}^{\text{inf}}$; we predict performance using generic scaling laws.

Each strategy operates in a configuration space that affects its resource requirements and potential gains: For QLoRA, $\Omega_{\text{QLoRA}} = [0, 1] \times \mathbb{N}^+$ is the spectrum of data proportion and discrete training iterations. The adaptation function maps a model to its fine-tuned version: $T_{\text{QLoRA}}^{\text{tr}} : f_{\eta,\phi} \times ([0, 1] \times \mathbb{N}^+) \to f_{\eta',\phi}$. For ICL, we control the number of shots $n$ and sequence length $\Omega_{\text{ICL}} = \{n \in \mathbb{N}^+ : C(n) \leq L_{\max}\}$ where $C(n) = L_{\text{query}} + \sum_{i=1}^n L_{\text{demo}_i} \leq L_{\max}$ represents the total sequence length. The ICL adaptation function modifies the input space: $T_{\text{ICL}}^{\text{inf}} : \mathcal{X} \times \Omega_{\text{ICL}} \to \mathcal{X}'$.

**Fine-tuning gain prediction.** For QLoRA fine-tuning, we develop an embedding-based prediction method. We use a language model $f_\theta$ in the model pool that has two key components: (1) a function $g_\eta : \mathbb{R}^{L \times d} \to \mathbb{R}^{L \times e}$, parameterized by $\eta$ that maps a sequence $\mathbf{x} = (x_1, \ldots, x_L)$ to a representation, where $d$ is the embedding dimension, and $e$ is the hidden dimension. It also has (2) a projection head $h_\phi : \mathbb{R}^e \to \mathbb{R}^{|\Sigma|}$, parameterized by $\phi$.

Inspired by (BehnamGhader et al., 2024), we first transform the traditional causal language model into a bidirectional embedding model. For input $\mathbf{x} = (x_1, \ldots, x_L)$, we compute: $z_t^{\text{bi}} = g_\eta^{\text{bi}}(x_1, x_2, \ldots, x_L)$ where $g_\eta^{\text{bi}}(\mathbf{x}) \in \mathbb{R}^{T \times e}$ produces contextualized representations. We use mean pooling to obtain a sequence embedding $e_\eta(\mathbf{x})$ used as input to the projector for fine-tuning performance estimator. Given the fine-tuning training data $D_{\text{train}}^{\text{FT}} = \{(\mathbf{x}_i, \mathbf{y}_i)\}_{i=1}^N$, we learn a lightweight *task-specific projector* $l_{\phi''} : \mathbb{R}^e \to \mathcal{Y}$ that maps sequence embeddings to the target space: $\hat{\mathbf{y}} = l_{\phi''}(e_\eta(\mathbf{x}))$ where $l_{\phi''}$ is a linear layer. Finally, to bridge the gap between projector predictions and actual fine-tuning performance, we use a calibration mechanism: $\hat{\pi}(T_{\text{QLoRA}}^{\text{tr}}(f_{\theta,\phi})) = a\pi_{\phi''} + b$ where $\pi_{\phi''}$ is the projector performance and $a, b \in \mathbb{R}$ are parameters learned from a small validation set (*e.g.*, 10% of full training data). This step ensures our predictions align with actual performance while maintaining computational efficiency.

Next, we describe costs for fine-tuning (and our predictor). The fine-tuning cost $c^{\text{FT}}$ depends on the number of tokens in the training set $N_{\text{train}}^{\text{FT}}$, the number of epochs $E$, batch size $B$, gradient accumulation step $G$, and the type of computational resources used. This factors in the total number of training steps, processing time per gradient update step $t_{\text{step}}$, and the hourly cost of compute resources $\gamma_{\text{compute}}$. We use token packing to optimize token usage and consider memory utilization $\psi_{\text{peak}}$ (the ratio of peak training memory occupation to total available memory), enabling fair comparison between prediction costs and full adaptation experiments. The total cost of fine-tuning is modeled as:

$$c^{\text{FT}} = E \times \frac{\text{pack}(N_{\text{train}}^{\text{FT}}, L_{\max})}{B \times G} \times t_{\text{step}} \times \gamma_{\text{compute}} \times N_{\text{compute}} \times \psi_{\text{peak}} + c_{\text{eval}}$$

where $\text{pack}(\cdot, \cdot)$ computes the number of effective sequences after optimal packing of training tokens $N_{\text{train}}^{\text{FT}}$ sequences subject to max sequence length $L_{\max}$ constraint. In terms of prediction, we derive the peak memory usage from the small validation set during performance calibration.

**Retrieval-augmented ICL gain prediction.** For retrieval-based ICL, our key insight is that retrieval-based ICL performance typically follows an exponential saturation curve requiring few measurements. Given measurements $\{(d_i, \pi_i)\}_{i=1}^{m}$, we fit the model: $\hat{\pi}(T_{\text{ICL}}^{\text{inf}}(f_{\eta,\phi})) = \alpha(1 - e^{-\beta d}) + \pi_0$, where $d$ is the shot count, and $(\alpha, \beta, \pi_0)$ capture saturation behavior. This allows us to predict performance at any count while requiring only a few initial points—as few as two.

For ICL, given query $\mathbf{x}$, we can estimate the cost: $c^{\text{ICL}}(d, \mathbf{x}) = c_{\text{token}}(\mathbb{E}[L_{\text{in}}] + \mathbb{E}[L_{\text{out}}]) \times d + c_{\text{token}}(\mathbf{x} + \mathbb{E}[L_{\text{out}}]) + c_{\text{eval}}$ where $\mathbb{E}[L_{\text{in}}]$ and $\mathbb{E}[L_{\text{out}}]$ are expected input/output lengths.

## 5 EXPERIMENTS

We conduct extensive experiments to validate COSMOS.

**Remark.** COSMOS is a *general analysis framework (abstraction)* for studying the strategy selection problem via predictive modeling (Sec. 3). It specifies *what* must be predicted, performance and cost for each model–strategy–configuration tuple, but is agnostic to *how* these predictions are obtained or which adaptation strategies are used. In our experiments, unless otherwise specified, COSMOS refers to its *reference instantiation* (Sec. 4), which instantiates the framework using two complementary predictor designs for QLoRA and retrieval-augmented ICL. Used together, these instantiations provide a concrete solution to the strategy selection problem by predicting both performance and cost across QLoRA and retrieval-augmented ICL.

> **Key Takeaway at a Glance**
>
> Optimizing training-time and inference-time strategies *jointly* can be more cost-effective than scaling them separately. COSMOS helps guide strategy selection by accurately and efficiently predicting both performance and cost, enabling practitioners to choose strategies *flexibly* based on their performance–cost tradeoffs.

Our evaluation aims to answer the following key questions:

- **Prediction Accuracy with Cost Efficiency** (Section 5.1): Can COSMOS effectively predict the optimal adaptation strategy? Our method achieves 92.72% cost reduction while maintaining high prediction fidelity (1.09% MAE) across tasks, strategy combinations, and cost regimes.
- **Robust Strategy-Specific Prediction Capabilities** (Section 5.2): How well does COSMOS predict the performance and cost of each combination? We demonstrate strong prediction capabilities for both performance gains and computational costs across multiple adaptation strategies and for general and specific tasks.
- **General vs. Tailored Predictors** (Section 5.3): What are the potential benefits of tailoring a predictor to the specific properties of an adaptation strategy?
- **Cost-effective Training-time and Test-time Scaling Synergies and Tradeoffs** (Section 5.4): What are the optimal performance-cost tradeoffs under different budgets when comparing train- vs. test-time adaptation? We provide critical insights into the efficiency of these approaches.
- **Strategy Space Expansion Benefits** (Section 5.5): How does broadening the adaptation strategy pool beyond simple model selection enhance the performance-cost tradeoffs in routing? We show augmenting model routing with our approach advances the Pareto frontier.

Table 1: Predicted vs. actual optimal strategies across tasks and cost regimes (over *55* strategy combinations of QLoRA and ICL). We achieve **substantial cost reduction across all levels** (92.72% average savings) while maintaining prediction fidelity (1.09% mean absolute error). Cost efficiency scales favorably with task size: larger tasks demonstrate greater absolute cost savings.

| Tasks | MMLU | | | Winogrande | | | ARC-Challenge | | | HellaSwag | | | FPB | | | FiQA-SA | | | Headline | | | Multifin EN | | | Avg. |
|---|---|---|---|---|---|---|---|---|---|---|---|---|---|---|---|---|---|---|---|---|---|---|---|---|---|
| Cost Level | L | M | H | L | M | H | L | M | H | L | M | H | L | M | H | L | M | H | L | M | H | L | M | H | |
| Pred. Acc (%) | 61.42 | 62.10 | 61.97 | 58.30 | 63.92 | 65.75 | 78.12 | 76.76 | 76.64 | 94.38 | 93.68 | 93.15 | 83.26 | 85.29 | 84.78 | 82.41 | 83.97 | 83.40 | 95.44 | 96.62 | 96.80 | 80.91 | 83.94 | 85.76 | - |
| Act. Acc (%) | 61.58 | 62.33 | 61.97 | 63.27 | 66.54 | 67.19 | 79.48 | 79.37 | 77.89 | 94.38 | 94.11 | 93.31 | 84.98 | 85.98 | 86.01 | 84.54 | 85.96 | 85.67 | 96.06 | 96.73 | 96.90 | 80.91 | 83.94 | 85.76 | - |
| MAE ↓ | 0.16 | 0.23 | 0.00 | 4.97 | 2.62 | 1.44 | 1.36 | 2.61 | 1.25 | 0.00 | 0.43 | 0.16 | 1.72 | 0.69 | 1.23 | 2.13 | 1.99 | 2.27 | 0.62 | 0.11 | 0.10 | 0.00 | 0.00 | 0.00 | **1.09** |
| Act. Cost ($) | 10.08 | 17.50 | 10.96 | 0.35 | 0.62 | 0.44 | 0.69 | 1.12 | 0.92 | 13.30 | 17.28 | 10.39 | 1.44 | 2.51 | 1.78 | 0.26 | 0.43 | 0.36 | 8.58 | 15.10 | 10.71 | 0.29 | 0.50 | 0.36 | - |
| Ours Cost ($) | 0.33 | 0.32 | 0.14 | 0.07 | 0.04 | 0.03 | 0.09 | 0.05 | 0.04 | 0.67 | 0.52 | 0.22 | 0.10 | 0.07 | 0.04 | 0.06 | 0.03 | 0.02 | 0.41 | 0.33 | 0.17 | 0.08 | 0.05 | 0.03 | - |
| CRR ↑ (%) | 96.68 | 98.17 | 98.71 | 80.91 | 93.99 | 94.31 | 87.35 | 95.35 | 96.12 | 94.99 | 96.99 | 97.90 | 92.88 | 97.33 | 97.81 | 76.48 | 92.77 | 93.38 | 95.19 | 97.81 | 98.44 | 70.92 | 89.84 | 90.95 | **92.72** |

- **Implications** (Section 5.6): How will COSMOS benefit real industrial deployment?

**Models, Tasks, and Metrics.** We use instruction-tuned versions of Gemma 2B (Gemma et al., 2024a) as a weaker model and Llama 3 8B (Dubey et al., 2024) as the stronger model. We evaluate COSMOS on a comprehensive suite of tasks spanning multiple domains. 1) *General Domain:* We evaluate on established benchmarks including Winogrande (Sakaguchi et al., 2021), ARC-Challenge (Clark et al., 2018), HellaSwag (Zellers et al., 2019) for commonsense reasoning, and MMLU (Hendrycks et al., 2020) for knowledge-based language understanding. 2) *Financial Domain:* We include FPB and FiQA-SA for sentiment analysis, and Headline and Multifin EN (Xie et al., 2023) for classification, representing domain-specific challenges. Detailed information is in the Appendix C.

We assess COSMOS via: 1) *Prediction Accuracy:* We measure performance and cost predictions using Mean Absolute Error (MAE): $\text{MAE} = \frac{1}{n} \sum_{i=1}^{n} |y_i - \hat{y}_i|$. Lower MAE indicates better prediction accuracy, and 2) *Cost Efficiency:* We quantify computational savings using Cost Reduction Ratio (CRR): $\text{CRR} = \frac{C_{\text{full}} - C_{\text{ours}}}{C_{\text{full}}} \times 100\%$, where $C_{\text{full}}$ represents total cost of evaluating all adaptation configurations, and $C_{\text{ours}}$ is the total cost of COSMOS to predict all those possibilities.

**Setup.** We evaluate COSMOS with QLoRA fine-tuning and retrieval-augmented ICL strategies. For QLoRA, the configuration space includes training iterations $\in \{4, 5, 6, 7, 8\}$ and data portions $\in \{0.1, \ldots, 1.0\}$ at 0.1 increments. For ICL, the number of demonstrations is $\{1, 2, 4, 8, 16\}$, constrained by max. sequence length (8,196 tokens). We use retrieval-augmented ICL using a BM25 Robertson et al. (2009) retriever to identify demonstrations. This yields 55 transformation combinations. All experiments are averaged over three random seeds. We partition the strategy space into three cost bands (low, medium, high) by uniformly dividing the range between min. and max. observed costs for each task. We evaluate COSMOS's ability to identify strategies that optimize the accuracy-cost tradeoff by maximizing predicted accuracy while minimizing the total monetary cost. Our score function is: $s(\pi, c) = \pi - \epsilon c / c_{\max}$ where $c_{\max}$ is the max. cost in that cost band, and $\epsilon$ is a small positive constant (*e.g.*, $\epsilon = 10^{-6}$); additional details are in Appendices D and E.

## 5.1 HOW WELL DOES COSMOS ADDRESS THE STRATEGY SELECTION PROBLEM?

Table 1 compares predicted vs. actual optimal strategies across 8 diverse tasks and cost regimes on Llama 3 8B. We report the actual accuracy of the predicted strategy (Pred. Acc), the best achievable accuracy (Act. Acc), and MAE (Eq. 5). For each level, we report costs of running all combinations (Act. Cost), running COSMOS (Ours Cost), and CRR. Our approach demonstrates efficiency-accuracy trade-offs, **achieving an average cost reduction of 92.72% while maintaining strong prediction fidelity: 1.09% MAE**. COSMOS exhibits two key scaling properties: (1) cost savings increase from low to high-cost ranges (improvement from 2.03% for MMLU to 20.03% for Multifin EN), i.e., better prediction capability in computationally intensive scenarios, and (2) cost efficiency improves with task scale, with larger tasks showing greater absolute savings (*e.g.*, MMLU: \$9.74-\$17.18, HellaSwag: \$12.64-\$16.76), i.e., COSMOS is increasingly advantageous as computational demands and task complexity grow.

We provide detailed comparisons with search-, training-, and scaling-law approaches in Appendix G. COSMOS consistently outperforms alternatives (achieves near-oracle accuracy while up to $49.1\times$ more cost-efficient, reduces prediction error by $3$–$5\times$ compared to other predictors). Results for an expanded model pool that further validate COSMOS's generalizability are in Appendix L.

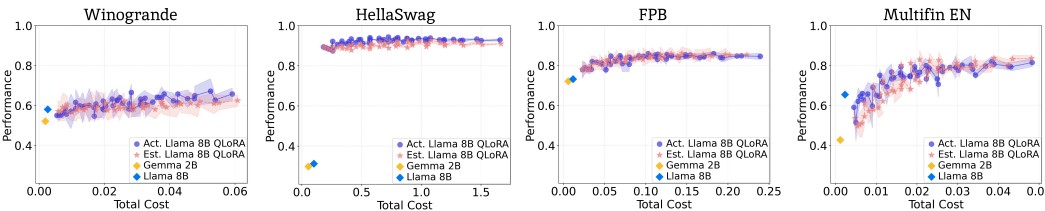

Figure 2: Actual (•) vs. predicted (⋆) performance-cost trajectories for Llama 3 8B QLoRA fine-tuning. Base models Gemma 2B (◆), and Llama 3 8B (◆) serve as reference points. The closer predicted (red) to the actual (purple) trajectories indicate better performance-cost prediction; we see consistent alignment between predicted and actual curves.

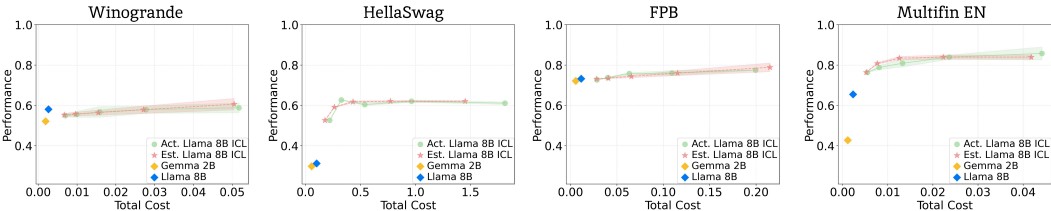

Figure 3: Predicted vs. actual performance-cost analysis for retrieval-based ICL. Each plot compares actual (•) vs. predicted (⋆) performance-cost trajectories for Llama 3 8B ICL. Base models Gemma 2B (◆), and Llama 8B (◆) serve as reference points. The consistent alignment between predicted and actual curves across all tasks demonstrates COSMOS's robust prediction capabilities.

## 5.2 STRATEGY-SPECIFIC ANALYSIS

Having established COSMOS's overall effectiveness in Section 5.1, we now present a detailed strategy-specific analysis of its prediction capabilities on all combinations.

Figure 2 shows the prediction accuracy for QLoRA fine-tuning. Each point is a fine-tuning configuration's performance-cost outcome (*e.g.*, training on 50% data for 5 iterations). The results reveal **strong prediction capabilities for all tasks**. For example, on FPB, COSMOS achieves high accuracy with MAE of 0.007 for both performance and cost predictions. We observe improved accuracy at higher computational budgets, where fine-tuning performance stabilizes. Even with limited training data and high-performance variance (low-cost scenario), COSMOS indicates if fine-tuning is worthwhile. These patterns persist across task domains. For ICL, Figure 3 shows the prediction accuracy. Each point is a ICL configuration's performance-cost outcome (*e.g.*, providing 8 demonstrations of input-output pairs in the query). The results reveal strong prediction capabilities across general-domain benchmarks and specialized financial tasks. For instance, on FiQA-SA, COSMOS achieves notably high accuracy with MAE of 0.003 and 0.001 for performance and cost predictions, respectively. Full results and analyses for all tasks and strategies are provided in Appendix H.

## 5.3 GENERAL VS. TAILORED PREDICTORS

Should we use COSMOS with a single universal predictor—or are there benefits to tailoring predictors? We compare the linear predictor against a generic scaling law approach. We use eight benchmarks with training iterations in $\{4, 5, 6, 7, 8\}$ and data portions in $\{0.1, 0.2, \ldots, 1.0\}$; We assume the same law—exponential saturation model—as used for ICL, and adopt the *most cost-efficient fitting strategy possible*, using the minimal data required (two data points). For each setting, we fit the model using data portions of 0.1 and 0.5,

| Prediction Method | MAE↓ | Total Cost↓ ($) |
|---|---|---|
| Scaling law-based | 1.96% | 13.91 |
| Linear predictor | 1.09% | 3.40 |
| Δ ↑ (Avg. on 8 tasks) | 44.60% | 75.57% |

Table 2: Prediction accuracy and cost for a *universal* approach vs. a *tailored* prediction strategy.

then extrapolate to other combinations. As shown in Table 2, *our embedding-augmented predictor significantly outperforms the scaling law based approach*, reducing prediction error by 44.60% while

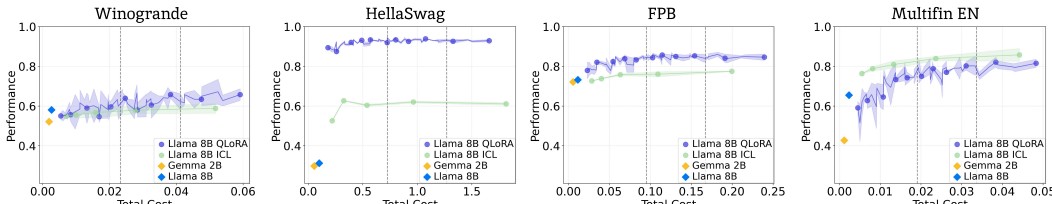

Figure 4: Actual QLoRA vs. ICL trajectories: performance-cost curves for QLoRA (●) and ICL (●) on Llama 3 8B, with Gemma 2B (◆), and Llama 3 8B (◆). Vertical dashed lines demarcate low, medium, and high-cost thresholds, determined by the min. and max. costs of both adaptation strategies. The shaded regions represent the standard deviation across 3 seeds for each configuration.

decreasing computational cost by 75.57%. These results suggests that tailoring prediction methods to specific strategies yields superior outcomes in both accuracy and efficiency.

## 5.4 COMBINING TRAINING- AND TEST-TIME STRATEGIES

We also conduct an analysis of the fundamental trade-offs between training-time and inference-time adaptation strategies by comparing QLoRA fine-tuning and retrieval-augmented ICL as illustrated in Figure 4. Full results can be found in Appendix J. We find: (1) **Non-linear Scaling Behaviors.** Both adaptation strategies exhibit diminishing returns with increased compute despite consistently outperforming the base model. Maximizing resources (shots for ICL or iterations/data for QLoRA) does not guarantee optimal performance. (2) **Stability-Performance Trade-offs.** QLoRA and ICL demonstrate distinct stability characteristics across different cost regimes. While QLoRA shows higher performance variance, particularly evident in Multifin EN where performance fluctuates significantly in low-cost settings ($\leq$\$0.019) before stabilizing at higher thresholds ($\geq$\$0.034), retrieval-augmented ICL maintains more consistent performance profiles, especially in resource-constrained scenarios. (3) **Resource-dependent Strategy Selection.** The optimal choice between fine-tuning and prompting depends on available resources. Fine-tuning typically achieves superior performance in medium to high-cost scenarios. ICL is a more reliable option in low-resource settings. (4) **Hybrid Strategy Benefits.** Strategically combining the approaches can achieve superior performance at lower costs. COSMOS can help produce this selection efficiently.

## 5.5 AUGMENTING ROUTING

Traditional model routing focuses on selecting from a pool of base models for each query. We show that expanding the routing space to include adaptation strategies can significantly enhance the performance-cost frontier. The benefits of this expanded strategy space are shown in Figure 5. Conventional routing selects between Gemma 2B and Llama 3 8B (old Pareto frontier). We establish a new frontier that substantially dominates the original (shaded red region).

Figure 5: Benefit of adaptation-augmented routing: *New Frontier* uses adaptation strategies (QLoRA, ICL). *Adaptation gains* are red-shaded area.

## 5.6 POTENTIAL IMPLICATIONS

Fine-tuning large language models (LLMs) at scale presents significant financial challenges. In Appendix K, we analyze these costs through a practical case study of fine-tuning GPT-4o, using OpenAI's current pricing structure. We obtain a rough approximation of cost savings of ~$939,830, bringing down the cost by a factor of 24.7x.

## 6  CONCLUSION

We formalized and studied the strategy selection problem–determining optimal combinations of models, adaptation approaches, and configurations while balancing performance and cost constraints. We introduced COSMOS, a framework that approaches this problem by predicting downstream performance and cost. Through two concrete instantiations and exhaustive experiments, we demonstrated that COSMOS can be realized effectively with minimal observations. COSMOS enables practitioners to navigate large model–strategy–configuration spaces efficiently and make flexible, informed, cost-aware decisions tailored to their performance–cost preferences.

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

**Appendix.** The appendix is organized as follows: We begin with limitations, societal impact, and disclosure of LLM usage in Appendix A, followed by a glossary table for key notations in the paper (Appendix B), details of the datasets (Appendix C) and experimental setup (Appendix D). Next, we describe our performance and cost prediction frameworks in Appendix E. We then present detailed evaluations of COSMOS (Appendices F–H), including extended results for Sections 5.1 and 5.2, comparisons against search-, training-, and scaling-law based methods (Appendix G), and in-depth strategy-specific analyses. We explore some theoretical bounds that predict the observed scaling laws, as well as a guarantee for model pruning in Hyperband-like pruning in Appendix I. We further provide extended results on strategy combinations (Appendix J) and discuss broader implications (Appendix K). Appendix L demonstrates generalizability across model families, Appendix M examines performance under limited data access, Appendix N studies transferability between models, and Appendix O evaluates applicability to additional adaptation strategies such as LoRA and dense retrieval. We present a concrete example illustrating how practitioners can conduct strategy selection using COSMOS's predicted metrics in Appendix P. We list examples of adaptation strategies in Appendix Q. Finally, we present a detailed discussion on recent scaling-law-based approaches in Appendix R.

## A  LIMITATIONS, SOCIETAL IMPACT AND USE OF LARGE LANGUAGE MODELS

**Limitations.**   This paper presents a novel framework for forecasting the performance and monetary cost of competing adaptation strategies and evaluates it across a broad set of language tasks and model families. We provide a theoretical understanding of the scaling regularities observed, however these results only provide upper bounds. Lower bounds, although more challenging and likely requiring assumptions on the model architecture, would enrich and validate the observed predictive models. Additionally, we note that our cost analysis relies on current model host (Together.ai and Vast.ai) token and GPU prices. However, the framework is modular: users can substitute region-specific prices or even carbon-intensity metrics with a single configuration change. Our empirical study mainly focuses on language models; we believe that exploring how our framework can extend to multimodal architectures and emerging agentic systems is an exciting direction for future work.

**Societal impact.**   This work activates accurate and low-overhead prediction of adaptation outcomes, which can improve access to advanced language model adaptations. By reducing the financial and environmental costs of trial-and-error experimentation, our framework lowers barriers for lower-resourced organizations. This enables broader participation in AI research, potentially accelerating innovation. The framework's ability to estimate performance-cost tradeoffs also supports more informed allocation of compute resources in industry and research. Since all experiments use established benchmarks for both training and retrieval, we do not anticipate any direct negative societal impacts arising from this work.

**The use of large language models.**   In this work, we mainly use LLMs to polish the writing of the drafted paper, rather than to generate the core content or sections.

## B  A GLOSSARY TABLE

We provide the key notations in Table 3.

## C  DATASETS DETAILS

Our evaluation spans both general-domain and domain-specific tasks, with dataset sizes varying from 380 to 21,570 examples (Table 4). This range allows us to systematically investigate model performance across both low-resource and data-rich scenarios.

For general-domain benchmarks (MMLU, Winogrande, HellaSwag, and ARC-Challenge), we maintain consistency with RouterBench (Hu et al., 2024) by adopting their prompting templates. This choice ensures comparability with model routing settings and eliminates potential performance variations due to prompt differences. For financial domain tasks (FPB, FiQA-SA, Headline, Multifin

Table 3: Glossary of key notations used in the strategy selection problem and COSMOS framework.

| Notation | Description |
|---|---|
| $\mathcal{D}$ | Set of downstream tasks |
| $D \in \mathcal{D}$ | A downstream task |
| $\mathcal{F} = \{f_1, \ldots, f_K\}$ | Model pool (candidate LLMs) |
| $f_k \in \mathcal{F}$ | A model mapping $\mathbf{x} \in \mathcal{X}$ to $\hat{\mathbf{y}} \in \mathcal{Y}$ |
| $\mathcal{T} = \{T_1, \ldots, T_J\}$ | Adaptation strategy pool (e.g., fine-tuning, ICL) |
| $T_j^\omega(f_k)$ | Model $f_k$ after applying strategy $T_j$ with configuration $\omega$ |
| $\Omega$ | Configuration space of parameters for adaptation strategies |
| $\omega \in \Omega$ | A specific configuration (e.g., data size, epochs, shots) |
| $\pi(\cdot)$ | Performance metric (e.g., accuracy) |
| $c(\cdot)$ | Cost function (adaptation + evaluation cost) |
| $s(\pi, c)$ | Score function capturing trade-off between performance and cost |
| $M_D$ | Strategy-selection indicator that outputs optimal model–strategy–config for task $D$ |
| $P_{j,k}(\omega)$ | Performance predictor for strategy $T_j$ on model $f_k$ under configuration $\omega$ |
| $C_{j,k}(\omega)$ | Cost predictor for strategy $T_j$ on model $f_k$ under configuration $\omega$ |
| $\hat{\pi}(T_j^\omega(f_k))$ | Predicted performance (via $P_{j,k}$) |
| $\hat{c}(T_j^\omega(f_k))$ | Predicted cost (via $C_{j,k}$) |
| $c_{\text{adapt}}$ | Cost of applying an adaptation strategy |
| $c_{\text{eval}}$ | Cost of evaluating model performance |
| $c_{\text{predict}}$ | Cost of training/calibrating predictors |

EN), we preserve the original dataset format as these typically include well-structured instructions and question-answer pairs.

Table 4: Number of training examples per task.

| Tasks | # of Train |
|---|---|
| MMLU | 9,809 |
| Winogrande | 886 |
| HellaSwag | 7,029 |
| ARC-Challenge | 1,029 |
| FPB | 3,100 |
| FiQA-SA | 750 |
| Headline | 21,570 |
| Multifin EN | 380 |

## D  EXPERIMENTAL DETAILS

**Training and retrieval setup.**   For datasets without predefined splits (general tasks), we implement a standard partition ratio of 70/10/20 for training, validation, and testing, respectively. For QLoRA fine-tuning (both full dataset training and prediction-time validation), we select data from the training set and evaluate performance on the test set. To account for data sampling variability when training with different data portions, we report average performance across three random seeds. For ICL, we employ BM25-based retrieval to dynamically select demonstrations from the training set for each test query. Our reported results represent averages across multiple demonstration orderings: the original retrieval order and two random permutations.

**Fine-tuning hyperparameters.**   For fine-tuning, we use a learning rate of 2e-4 (following Dettmers et al. (2024)), batch size of 1, maximum sequence length of 512, gradient accumulation steps of 2, warmup ratio of 0.03, maximum gradient norm of 0.3, LoRA alpha of 32, LoRA dropout of 0.05, LoRA rank of 64. To standardize measurements across varying cluster loads, we assume an average step time of 1.09 seconds, which corresponds to the mean processing time on uncontested GPU resources.

**Hardware and software.** We conduct experiments using 15 NVIDIA A100-PCIE-40GB and 1 NVIDIA A100-SXM4-40GB GPUs, with Python 3.10, PyTorch 2.5.1, and Transformers 4.45.2.

**Cost assumptions.** We use an hourly rate of \$1/h for A100-40GB based on Vast.ai pricing listed on a GPU comparison website[1] for computing-based cost estimation. For token-based fine-tuning costs, we extrapolate from Together.ai[2]'s online cost calculator and fit Llama 3 8B's pricing using a power law to calculate the total cost per epoch $c_{\text{epoch}} = a \times (N_{\text{token}})^b$, excluding their \$5 minimum cost requirement, where $a \approx 8.69e - 7$, $b \approx 0.956$. For inference cost, we utilize Together.ai's pricing of \$0.2 for Llama 3 8B per million tokens and \$0.1 for Gemma 2B per million tokens.

**Cost of validation.** Validating COSMOS involved a very large number of experiments, which would be computationally prohibitive if conducted with full fine-tuning. Across 5 models, 8 benchmarks, 3 seeds, 10 data portions, 5 iterations, and 5 LoRA rank values, we tuned approximately 7,100 checkpoints. Storing full fine-tuned checkpoints (average 26.5GB each) would require ∼187TB of storage. In contrast, QLoRA checkpoints are only ∼1.2GB each, reducing the total storage requirement to ∼8.3TB—a practical scale for systematic study.

In terms of compute, we have expended over 10,000 GPU hours to obtain the QLoRA results. Performing equivalent full fine-tuning would require at least $16\times$ more GPU memory and compute (Dettmers et al., 2024), totaling over 162,000 GPU hours. At an hourly rate of \$1 for A100-40GB GPUs, this translates to an estimated cost of \$162,500, which is beyond our means. These figures underscore the computational difficulty of exhaustive adaptation studies, and motivate the need for predictive frameworks such as COSMOS.

# E  PERFORMANCE AND COST PREDICTION

## E.1  FINE-TUNING

Our prediction framework employs task-specific approaches based on complexity. For complex tasks (MMLU, HellaSwag, Winogrande, ARC-Challenge), we implement a contrastive learning approach using a lightweight linear projector. This model is trained using correct answers as positive examples and incorrect options as negative samples, with the following configuration: batch size of 8, maximum sequence length of 512, learning rate of 1e-6, and temperature of 0.07. The architecture consists of layer normalization followed by a linear projection layer, with model selection based on peak test accuracy over 300 iterations.

For financial domain tasks, we adopt a more streamlined approach, utilizing a single linear layer trained with cross-entropy loss on model-generated embeddings. This simplified architecture achieves rapid training convergence on CPU, resulting in negligible proxy model training costs.

To ensure robust performance predictions across different data regimes, we employ a calibration process using a minimal subset of training data—either 200 examples or 10% of the training set, whichever is larger. This one-time calibration yields scaling factors applicable across all data portion configurations (0.1 through 1.0) per epoch.

**Cost prediction.** The training cost is calculated by multiplying GPU usage duration and peak memory utilization by the compute price. To standardize measurements across varying cluster loads, we assume a fixed processing time of 0.0009 seconds per data point per epoch on an idle NVIDIA A100-PCIE-40GB, which corresponds to the mean processing time on uncontested GPU resources.

The primary prediction cost stems from validation set training. This explains the varying cost reduction rates across tasks–achieving up to 98.71% reduction for MMLU in high-budget scenarios, while showing lower reductions for smaller datasets like Multifin EN (380 total points). Since validation costs are amortized across all strategies using the same scaling factor (currently 10 in our experiments), increasing the number of strategies or configurations would further improve cost efficiency.

---

[1] https://cloud-gpus.com/
[2] https://www.together.ai/pricing

## E.2 Retrieval-augmented ICL

We discover that retrieval-augmented ICL performance can be effectively predicted using minimal data (as few as 2 samples). Building on this finding, we fit an exponential saturation function (detailed in Section 4) using performance measurements from 1-shot and 8-shot settings. For the baseline performance $\pi_0$, we select the lower value between zero-shot performance and 1-shot results.

Cost estimation for ICL leverages the average input and output lengths observed in the training set for efficiency.

# F  A Detailed Performance Analysis of COSMOS

Table 5: Comprehensive evaluation of our strategy prediction framework across 8 diverse tasks under different cost regimes. Results compare predicted vs. actual optimal strategies across low (L), medium (M), and high (H) cost settings, evaluating *55* combinations of QLoRA and ICL techniques. The analysis encompasses multiple accuracy metrics (predicted, actual, mean, extremal values, and range averages) and their corresponding cost measurements, demonstrating our method's effectiveness in identifying optimal strategies while maintaining performance across varying computational budgets.

| Tasks | MMLU | | | Winogrande | | | ARC-Challenge | | | HellaSwag | | | FPB | | | FiQA-SA | | | Headline | | | Multifin EN | | |
|---|---|---|---|---|---|---|---|---|---|---|---|---|---|---|---|---|---|---|---|---|---|---|---|---|
| Cost Level | L | M | H | L | M | H | L | M | H | L | M | H | L | M | H | L | M | H | L | M | H | L | M | H |
| Min Acc (%) | 55.91 | 57.32 | 61.14 | 54.64 | 57.91 | 58.82 | 73.70 | 75.96 | 75.06 | 52.56 | 62.02 | 61.09 | 72.65 | 76.05 | 77.49 | 74.18 | 78.16 | 80.85 | 72.91 | 74.90 | 70.52 | 51.52 | 70.68 | 75.27 |
| Max Acc (%) | 61.58 | 62.33 | 61.97 | 63.27 | 66.54 | 67.19 | 79.48 | 79.37 | 77.89 | 94.38 | 94.11 | 93.31 | 84.98 | 85.98 | 86.01 | 84.54 | 85.96 | 85.67 | 96.06 | 96.73 | 96.90 | 80.91 | 83.94 | 85.76 |
| Avg Acc (%) | 59.96 | 61.14 | 61.60 | 58.22 | 62.16 | 63.73 | 77.44 | 77.62 | 76.70 | 88.35 | 91.36 | 88.34 | 80.70 | 84.29 | 83.89 | 80.14 | 82.17 | 83.65 | 91.51 | 95.07 | 93.72 | 70.05 | 77.86 | 80.27 |
| Act. Acc (%) | 61.58 | 62.33 | 61.97 | 63.27 | 66.54 | 67.19 | 79.48 | 79.37 | 77.89 | 94.38 | 94.11 | 93.31 | 84.98 | 85.98 | 86.01 | 84.54 | 85.96 | 85.67 | 96.06 | 96.73 | 96.90 | 80.91 | 83.94 | 85.76 |
| R.Avg Acc (%) | 58.75 | 59.83 | 61.55 | 58.95 | 62.22 | 63.01 | 76.59 | 77.62 | 76.47 | 73.47 | 78.07 | 77.20 | 78.81 | 81.01 | 81.75 | 79.36 | 82.06 | 83.65 | 84.49 | 85.81 | 83.71 | 66.21 | 77.31 | 80.52 |
| Pred Acc (%) | 61.42 | 62.10 | 61.97 | 58.30 | 63.92 | 65.75 | 78.12 | 76.76 | 76.64 | 94.38 | 93.68 | 93.15 | 83.26 | 85.29 | 84.78 | 82.41 | 83.97 | 83.40 | 95.44 | 96.62 | 96.80 | 80.91 | 83.94 | 85.76 |
| Min Cost ($) | 0.163 | 0.677 | 1.189 | 0.005 | 0.024 | 0.042 | 0.011 | 0.047 | 0.079 | 0.181 | 0.758 | 1.270 | 0.023 | 0.097 | 0.171 | 0.004 | 0.018 | 0.031 | 0.135 | 0.584 | 1.034 | 0.004 | 0.019 | 0.034 |
| Max Cost ($) | 0.639 | 1.152 | 1.663 | 0.023 | 0.041 | 0.059 | 0.044 | 0.072 | 0.110 | 0.725 | 1.192 | 1.815 | 0.092 | 0.166 | 0.239 | 0.017 | 0.028 | 0.044 | 0.550 | 0.999 | 1.448 | 0.018 | 0.033 | 0.048 |
| Avg Cost ($) | 0.388 | 0.875 | 1.370 | 0.013 | 0.031 | 0.049 | 0.026 | 0.059 | 0.092 | 0.443 | 0.960 | 1.484 | 0.055 | 0.125 | 0.197 | 0.010 | 0.023 | 0.036 | 0.330 | 0.755 | 1.190 | 0.011 | 0.025 | 0.040 |
| Act. Cost ($) | 10.076 | 17.501 | 10.963 | 0.351 | 0.621 | 0.443 | 0.688 | 1.116 | 0.920 | 13.305 | 17.282 | 10.385 | 1.440 | 2.506 | 1.775 | 0.261 | 0.430 | 0.357 | 8.580 | 15.097 | 10.713 | 0.287 | 0.504 | 0.360 |
| R.Avg Cost ($) | 0.401 | 0.914 | 1.426 | 0.014 | 0.033 | 0.051 | 0.027 | 0.059 | 0.095 | 0.453 | 0.975 | 1.543 | 0.057 | 0.131 | 0.205 | 0.010 | 0.023 | 0.036 | 0.342 | 0.792 | 1.241 | 0.011 | 0.026 | 0.041 |
| Ours Cost ($) | 0.335 | 0.320 | 0.142 | 0.067 | 0.037 | 0.025 | 0.087 | 0.052 | 0.036 | 0.666 | 0.520 | 0.218 | 0.103 | 0.067 | 0.039 | 0.061 | 0.031 | 0.024 | 0.413 | 0.331 | 0.167 | 0.083 | 0.051 | 0.033 |

In Section 5.1, we presented the comparison of predicted versus actual optimal strategies. Here, we provide a comprehensive analysis through Table 5, which details multiple performance dimensions across different cost regimes. For each task and cost level (Low/Medium/High), we report both accuracy and cost metrics. The accuracy metrics include our strategy's predicted performance (Pred Acc), the actual optimal strategy's performance (Act Acc), and statistical measures across all strategies (minimum, maximum, and average accuracy). This allows us to evaluate our method's effectiveness from multiple angles. To quantify the cost-efficiency of our approach, we compare three key metrics: (1) the total cost of exhaustively evaluating all 55 strategy combinations (Act Total Cost), (2) our method's prediction cost (Ours Total Cost), and (3) baseline costs from random strategy selection within each cost band (Range Avg Cost).

To further illustrate our method's effectiveness, consider the HellaSwag task under the high-cost regime: our predictor achieves 93.15% accuracy at $0.218, significantly outperforming random strategy selection which yields 77.2% accuracy at $1.543. This demonstrates that our approach not only maintains near-optimal performance but also reduces computational costs by over 7x compared to random selection within the target cost range.

We also present a concrete example on how to select the optimal strategy based on predicted metrics given by COSMOS in Appendix P.

# G  Comparison With Other Search-, Training-, and Scaling-law based Methods

## G.1  Comparison with Search- and Training-based Baselines

To contextualize the effectiveness of COSMOS, we compare it against several representative baselines that can be adapted for strategy selection. While we are not aware of any prior framework that jointly

Table 6: COSMOS significantly performs better than baseline methods in both performance prediction accuracy and cost efficiency, which achieves near-oracle accuracy (99.3%-100%) while reducing computational costs by up to 49.1x across all budget levels.

| Cost Level | Methods | Acc. | Prediction Cost↓ ($) |
|---|---|---|---|
| Low | Oracle | 0.944 | - |
| | RS-CV | 0.933 | 1.474 |
| | Hyperband (Yu & Zhu, 2020) | 0.923 | 1.391 |
| | FrugalGPT (Chen et al., 2023) | 0.940 | 12.580 |
| | RouterBench (Hu et al., 2024) | 0.919 | 8.580 |
| | COSMOS (Ours) | **0.944** | **0.666** |
| Medium | Oracle | 0.941 | - |
| | RS-CV | 0.936 | 3.621 |
| | Hyperband (Yu & Zhu, 2020) | 0.934 | 3.999 |
| | FrugalGPT (Chen et al., 2023) | 0.940 | 5.760 |
| | RouterBench (Hu et al., 2024) | 0.926 | 15.097 |
| | COSMOS (Ours) | **0.937** | **0.520** |
| High | Oracle | 0.933 | - |
| | RS-CV | 0.927 | 5.990 |
| | Hyperband (Yu & Zhu, 2020) | 0.931 | 5.914 |
| | FrugalGPT (Chen et al., 2023) | 0.931 | 5.446 |
| | RouterBench (Hu et al., 2024) | 0.929 | 10.713 |
| | COSMOS (Ours) | **0.932** | **0.218** |

predicts both performance and cost across multi-model, multi-strategy (training- and test-time), multi-configuration, and multi-task settings, training a classifier or search-based paradigms can be applied to the strategy selection problem. Specifically, we consider Random Search with Cross-Validation (RS-CV), Hyperband (Li et al., 2017), FrugalGPT (Chen et al., 2023), and RouterBench (Hu et al., 2024). These methods rely on repeated trials or search heuristics to approximate performance, but they cannot simultaneously model cost and accuracy for direct predictive selection.

**Experimental Setup.** We conduct experiments on the HellaSwag benchmark and evaluate two complementary metrics: (i) **prediction accuracy**, measured as agreement with the oracle (optimal performance under exhaustive search), and (ii) **cost efficiency**, measured as the computational expenditure required to obtain predictions. This setup highlights not only whether predictions are correct, but also whether they are obtained efficiently.

**Results.** Table 6 shows that COSMOS consistently delivers near-oracle accuracy while being substantially more cost efficient. In the low-, medium-, and high-budget regimes, COSMOS achieves 100%, 99.3%, and 99.9% of oracle accuracy, respectively. At the same time, it reduces costs by $2.2\times$, $7.0\times$, and $24.9\times$ compared to the strongest baseline in each setting. For instance, in the high-cost regime, COSMOS attains 0.932 accuracy versus the oracle's 0.933, but requires only \$0.218—$27.1\times$ cheaper than Hyperband.

**Discussion.** These results illustrate a paradigm shift from traditional search-based methods, which must run full experiments to observe actual performance and cost. By contrast, COSMOS directly predicts these outcomes, eliminating the need for exhaustive trials or extensive validation. This predictive capability enables practitioners to make informed, cost-aware decisions with minimal overhead, a particularly important advantage in resource-constrained environments.

### G.2 COMPARISON WITH OTHER SCALING-LAW PREDICTORS

Recent work has investigated scaling-law based predictors for estimating model performance, often using FLOPs, parameter counts, token budgets, or low-level skills as proxies (Kaplan et al., 2020; Owen, 2024; Ruan et al., 2024; Polo et al., 2024). While these methods can capture coarse performance trends, they are not designed for multi-strategy or cost-sensitive selection and fail to

capture the real training nuance in the real-world deployment. We compare COSMOS with several representative scaling-law methods on four reasoning benchmarks. We report mean absolute error (MAE, in percentage points) between predicted and actual performance, lower is better.

Table 7: Comparison of COSMOS with recent scaling-law predictors. Results are mean absolute error (MAE, in percentage points). COSMOS achieves a 3–5× reduction in error across all tasks.

| Method | MMLU | ARC-Challenge | HellaSwag | Winogrande |
|---|---|---|---|---|
| FLOPs-based (Owen, 2024) | 8.0 | 4.6 | 5.1 | 3.7 |
| Size & Tokens (Kaplan et al., 2020) | 7.1 | 3.2 | 3.8 | 2.6 |
| PCA+FLOPs (Ruan et al., 2024) | 9.3 | 3.8 | 5.4 | 2.7 |
| Sloth (Polo et al., 2024) | 6.2 | 3.8 | 5.7 | 3.3 |
| COSMOS (Ours) | **1.5** | **1.3** | **1.8** | **2.2** |

As shown in Table 7, COSMOS consistently outperforms scaling-law predictors, achieving 3–5× lower MAE across MMLU, ARC-Challenge, HellaSwag, and Winogrande. This highlights that COSMOS not only extends beyond parametric scaling proxies but also achieves substantially more accurate performance prediction, which is crucial for reliable cost-aware strategy selection.

## H  DETAILED ANALYSIS OF PREDICTION CAPABILITIES OF COSMOS

### H.1  FULL RESULTS FOR STRATEGY-SPECIFIC ANALYSIS

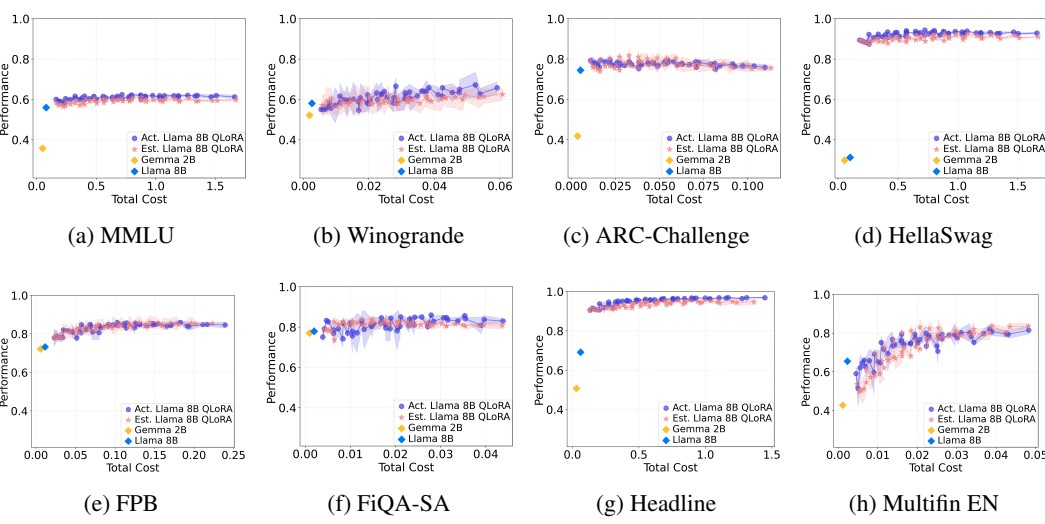

(a) MMLU  (b) Winogrande  (c) ARC-Challenge  (d) HellaSwag

(e) FPB  (f) FiQA-SA  (g) Headline  (h) Multifin EN

Figure 6: Predicted vs. actual performance-cost analysis for QLoRA fine-tuning across eight diverse tasks. Each plot compares actual (•) vs. predicted (⋆) performance-cost trajectories for Llama 3 8B QLoRA fine-tuning. Base models Gemma 2B (◆), and Llama 3 8B (◆) serve as reference points. The closer predicted performance-cost trajectories (red) to the actual (purple) trajectories indicates better performance-cost prediction. The results show a consistent alignment between predicted and actual curves across both general and domain-specific tasks. This demonstrates COSMOS's robust prediction capabilities.

Following our analysis in Section 5.2, we present comprehensive performance–cost trajectories for all eight tasks in Figure 6 and 7, examining QLoRA fine-tuning and retrieval-augmented ICL, respectively. The strong alignment between predicted and actual performance trajectories across all tasks and strategies validates our method's robustness. Our framework demonstrates particular strength in capturing complex performance dynamics–not only predicting standard improvement curves, but also accurately forecasting non-monotonic patterns, such as the performance degradation observed in the ARC-Challenge task as the computational budget increases. This ability to capture both positive and negative performance trends further substantiates the generalizability of our prediction framework.

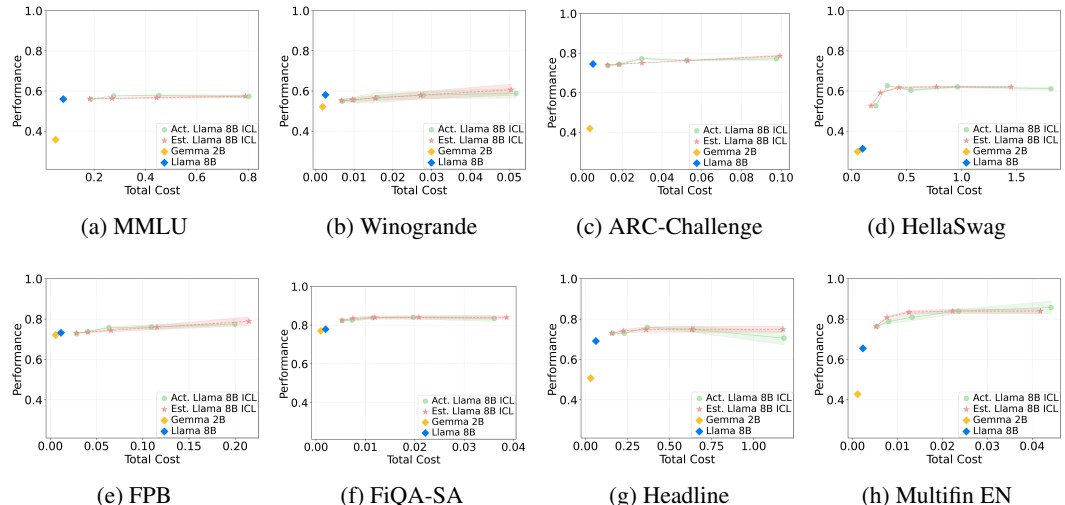

(a) MMLU     (b) Winogrande     (c) ARC-Challenge     (d) HellaSwag

(e) FPB     (f) FiQA-SA     (g) Headline     (h) Multifin EN

Figure 7: Predicted vs. actual performance-cost analysis for retrieval-based ICL across eight diverse tasks. Each plot compares actual (●) vs. predicted (⋆) performance-cost trajectories for Llama 3 8B ICL. Base models Gemma 2B (◆), and Llama 8B (◆) serve as reference points. The consistent alignment between predicted and actual curves across both general and domain-specific tasks demonstrates COSMOS's robust prediction capabilities.

## H.2 A CLOSER LOOK AT COSMOS'S PREDICTION ABILITY FOR FINE-TUNING

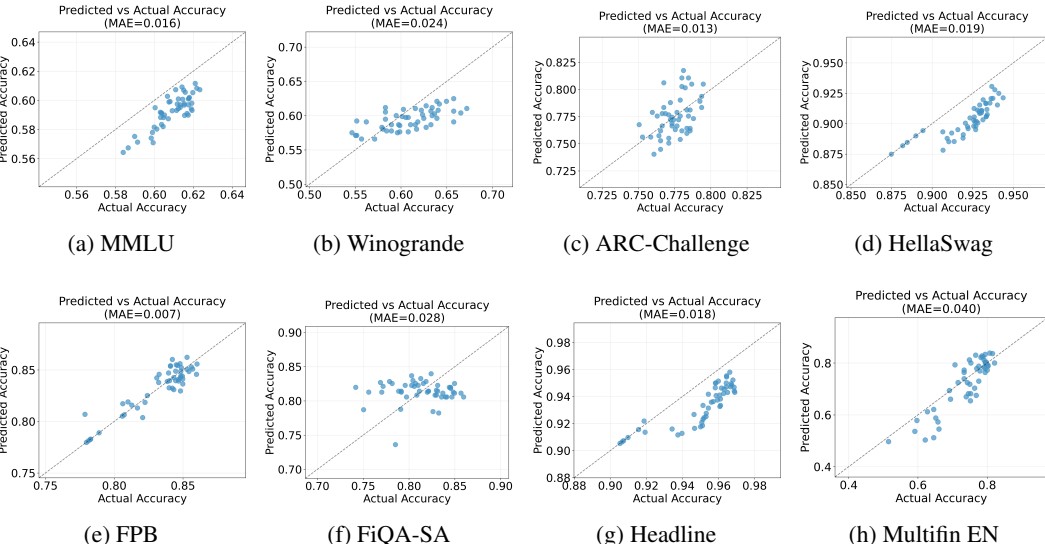

(a) MMLU     (b) Winogrande     (c) ARC-Challenge     (d) HellaSwag

(e) FPB     (f) FiQA-SA     (g) Headline     (h) Multifin EN

Figure 8: Scatter plots comparing predicted vs. actual performance (accuracy) for QLoRA fine-tuning across eight diverse tasks Axes are strategically zoomed to reveal fine-grained prediction details, with points closer to the diagonal indicating higher prediction accuracy. For example, in Headline, a ● at (0.92, 0.915) shows that for a specific configuration (training data size and iteration count), COSMOS predicts 0.915 accuracy while the actual performance achieves 0.92. The tight clustering around the diagonal across both general domain (a-d) and financial domain (e-h) tasks demonstrates COSMOS's robust prediction capabilities.

**Performance prediction.** Figure 8 provides a detailed analysis of COSMOS's prediction accuracy on fine-tuning across eight tasks. For each task, we plot predicted versus actual performance with axes deliberately zoomed to highlight prediction granularity. Each scatter point represents a specific

QLoRA fine-tuning configuration, with proximity to the diagonal indicating prediction accuracy. The Mean Absolute Error (MAE) ranges from 0.007 (FPB) to 0.040 (Multifin EN), with most tasks showing MAE below 0.02, demonstrating remarkable precision. The framework exhibits consistent performance across both general-domain benchmarks (MMLU: 0.016, Winogrande: 0.024, ARC-Challenge: 0.013, HellaSwag: 0.019) and financial tasks (FPB: 0.007, FiQA-SA: 0.028, Headline: 0.018, Multifin EN: 0.040). The tight clustering around the diagonal, particularly evident in the zoomed visualization, underscores our method's robust predictive capabilities regardless of task domain or performance level.

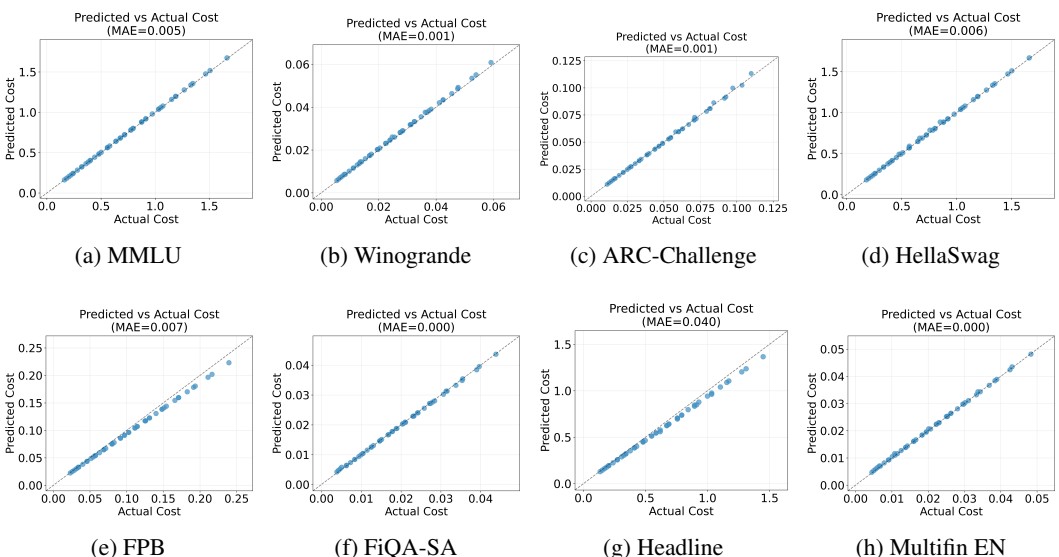

(a) MMLU  (b) Winogrande  (c) ARC-Challenge  (d) HellaSwag

(e) FPB  (f) FiQA-SA  (g) Headline  (h) Multifin EN

Figure 9: Scatter plots of predicted vs. actual cost for QLoRA fine-tuning across eight diverse tasks. The near-perfect diagonal alignment and low MAE values (0.000-0.007) demonstrate precise cost prediction capabilities across both resource-intensive (a-d) and lightweight tasks (e-h).

**Cost prediction.**   Figure 9 demonstrates our framework's cost prediction capabilities for QLoRA fine-tuning across eight tasks. The scatter plots reveal near-perfect diagonal alignment with remarkably low MAE (0.000-0.007) across all tasks, from resource-intensive benchmarks like MMLU to lightweight tasks like FiQA-SA. This consistency across varying cost scales validates our framework's robust cost estimation capabilities.

### H.3   A CLOSER LOOK AT COSMOS'S PREDICTION ABILITY FOR RETRIEVAL-AUGMENTED IN-CONTEXT LEARNING

**Performance prediction.**   Figure 10 demonstrates our framework's prediction accuracy for retrieval-augmented ICL across eight tasks. With deliberately zoomed axes to highlight prediction granularity, the scatter plots reveal strong performance with MAE ranging from 0.003 (FiQA-SA) to 0.013 (Multifin EN and Headline). The framework maintains consistent accuracy across both general-domain tasks (MMLU: 0.007, Winogrande: 0.005, ARC-Challenge: 0.008, HellaSwag: 0.012) and financial tasks (FPB: 0.007, FiQA-SA: 0.003, Headline: 0.013, Multifin EN: 0.013), with an average MAE of 0.0085. This low average deviation of 0.85% from actual performance validates our framework's robust prediction capabilities across diverse domains.

**Cost prediction.**   Figure 11 demonstrates our framework's cost prediction capabilities for ICL across eight tasks. Most tasks show excellent prediction accuracy with MAE ranging from 0.001 (Winogrande, ARC-Challenge, FiQA-SA, Multifin EN) to 0.009 (MMLU), with HellaSwag being the only outlier (MAE=0.154). This higher deviation in HellaSwag stems from our design choice to use training set averages for sequence length estimation instead of observed samples during performance model fitting, prioritizing efficiency over perfect accuracy. While this approximation is typically

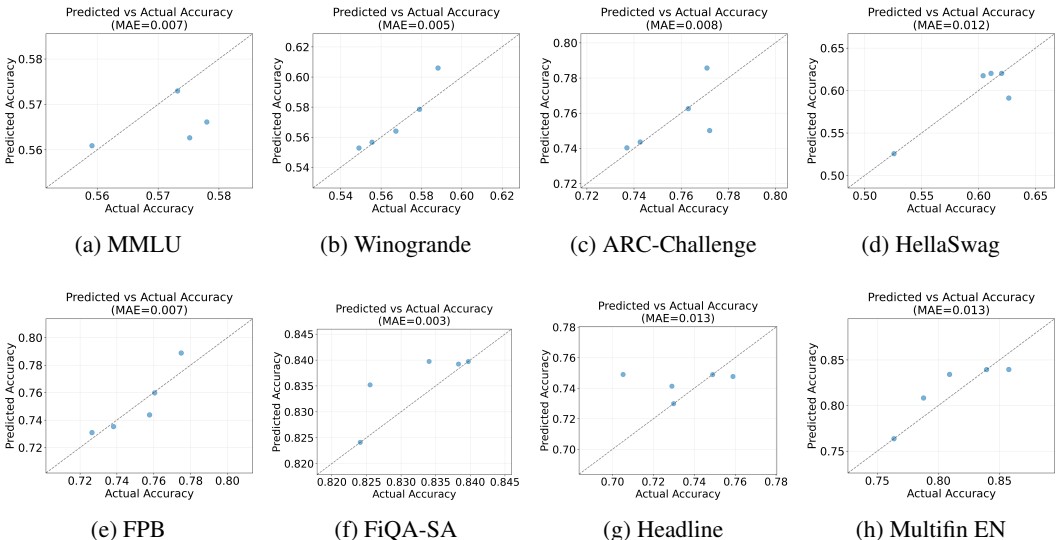

Figure 10: Scatter plots comparing predicted vs. actual accuracy for ICL across eight diverse tasks. Axes are zoomed to highlight fine-grained prediction details, with an average MAE of 0.85% demonstrating high prediction fidelity. The consistent performance across both general domain (a-d) and financial domain (e-h) tasks validates our framework's robust prediction capabilities.

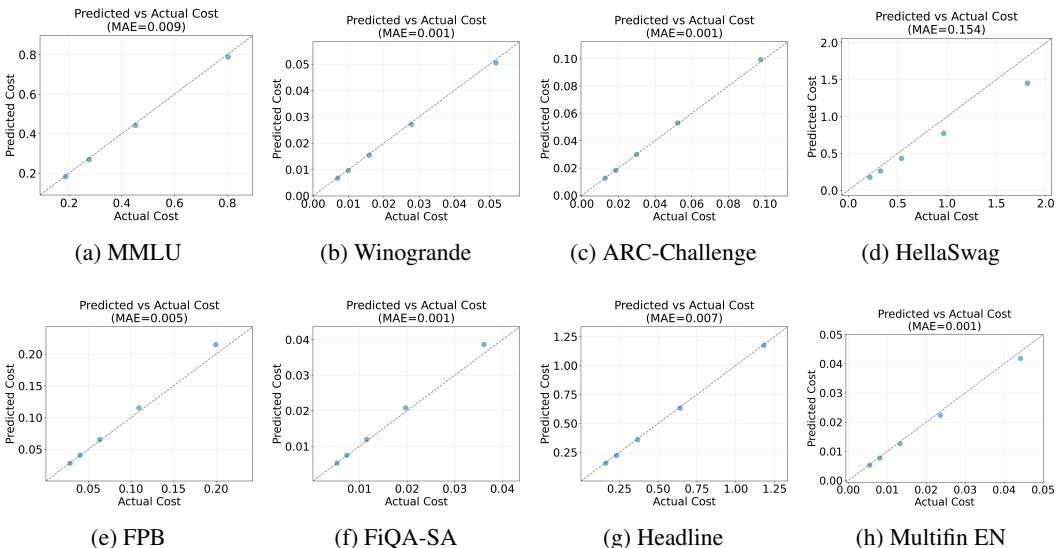

Figure 11: Scatter plots of predicted vs. actual cost for ICL across eight diverse tasks.

sufficient for cost estimation, prediction accuracy could be trivially improved by using observed sample lengths when higher precision is needed.

# I    THEORETICAL RESULTS

We investigate some reasons to expect COSMOS's performance predictions to be accurate, as well as the added benefit of pruning further hyper parameter search.

## I.1 NOTATION

Let $f_\theta$ be our model which takes examples $x$ from some vocabulary $\mathcal{V}$ and generates an output $y$ in $V \subset \mathcal{V}$. Also, let $f_\theta = e_\theta \circ h_\theta$, where $h_\theta$ is a linear transformation (the learned head in linear probing) while $e_\theta$ is the token embedding for the model. We are interested in 2 different additionally learned parameters, where $\phi'$ are the model parameters where $e_\theta$ is left unchanged and $\phi$ are the parameters when fine-tuning the entire model. Specifically, we are interested in $\pi(\phi')$ and $\pi(\phi)$, where

$$\pi(\phi) = P_{(x,y)\sim\mathcal{D}}(f_\phi(x) = y).$$

## I.2 QLoRA PERFORMANCE

Let the following be the optimal parameters for each of these settings:

$$\phi_* = \arg\min_\phi \pi(\phi),$$

$$\phi'_* = \arg\min_{\phi'} \pi(\phi'),$$

where the second can only update the parameters of the final layer $h$. We require some further assumptions about the training dynamics:

**Assumption I.1.** The error $P(f_{\phi_*} \neq f_\phi)$ is bounded by a small function $\epsilon(C)$ that depends on the cost needed to train the model that vanishes to 0.

**Theorem I.2.** *For original parameters $\theta$ and a linear-probing dataset size $n$, the result of of linear-probing ($\phi'$) fine-tuning ($\phi$) satisfy the following bound:*

$$\pi(\phi) \leq \pi(\phi') + b(C) + O(1/\sqrt{n}),$$

*where $b$ is a function constant for any fixed cost of training.*

*Proof.* This follows from a cascading union bound.

$$1 - \pi(\phi') = P(y \neq f_{\phi'}).$$
$$\leq P(y \neq f_\phi) + P(f_\phi \neq f_{\phi_*}) + P(f_{\phi_*} \neq f_{\phi'_*}) + P(f_{\phi'_*} \neq f_{\phi'}).$$
$$= 1 - \pi(\phi) + P(f_\phi \neq f_{\phi_*}) + P(f_{\phi_*} \neq f_{\phi'_*}) + P(f_{\phi'_*} \neq f_{\phi'}).$$

Thus,

$$\pi(\phi) \leq \pi(\phi') + P(f_\phi \neq f_{\phi_*}) + P(f_{\phi_*} \neq f_{\phi'_*}) + P(f_{\phi'_*} \neq f_{\phi'}).$$

From the assumption about $P(f_{\phi_*} \neq f_\phi)$, we can bound that term by $\epsilon(C)$. Also, since $P(f_{\phi_*} \neq f_{\phi'_*})$ is some constant that does not depend on the training dynamics, we can define

$$b(C) = \epsilon(C) + P(f_{\phi_*} \neq f_{\phi'_*}).$$

Lastly, $P(f_{\phi'_*} \neq f_{\phi'})$ excess risk of training a linear classifier, which is $O(1/\sqrt{n})$ following VC theory. In all, we have

$$\pi(\phi) \leq \pi(\phi') + b(C) + O(1/\sqrt{n}).$$

$\square$

## I.3 ICL PERFORMANCE

The stated scaling law for ICL performance with $d$ examples is $\hat{\pi} = \alpha(1 - e^{-\beta d}) + \pi_0$, for learned parameters $\alpha, \beta, \pi_0$. We show that, assuming the model $f$ acts like a Bayesian predictor based on the examples in its context, that it is possible to predict this scaling law for a collection of tasks.

First, let $\mathcal{T}$ be a collection of tasks, where a task is a probability distribution over the vocabulary $\mathcal{V}$, denoted as $P(\cdot|T)$ for $T \in \mathcal{T}$. Also assume that the samples generated in the context $D$ are i.i.d. from some chosen task, i.e.

$$P(D|T) = \prod_{i=1}^{n} P(D_i|T).$$

Now, when trying to generate a new token $v$, we have that $v$ and $D$ are conditionally independent given $T$, so

$$P(v|T, D) = P(v|T).$$

We need the following assumption and parameter $\gamma$, which says that the chosen task and context is sufficiently different than any of the other tasks.

**Assumption I.3.** Denote $\overline{\overline{T}} = \mathcal{T} \setminus \{T\}$. For any task $T$, $D$ drawn from $T$, there exists some $\gamma > 1$ such that $P(D_i|T) > \gamma P(D_i|\overline{T})$.

**Theorem I.4.** *For a task $T \in \mathcal{T}$, a context $D$ drawn from $T$ of length $d >> 1$, then*

$$P(v|D) \geq P(v|T) - \gamma^{-d} \frac{(P(v|T) - P(v|\overline{T}, D))P(\overline{T})}{P(T)}.$$

*Under the following definitions,*

$$\alpha = \frac{(P(v|T) - P(v|\overline{T}, D))P(\overline{T})}{P(T)},$$

$$\beta = \log \gamma,$$

$$\pi_0 = P(a|T) - \alpha,$$

*then*

$$P(v|D) \geq \alpha(1 - e^{\beta d}) + \pi_0.$$

*Proof.* Following the law of total probability and conditional independence,

$$P(v|D) = P(v|T)P(T|D) + P(v|\overline{T}, D)P(\overline{T}, D),$$
$$= P(v|T)(1 - P(\overline{T}, D)) + P(v|\overline{T}, D)P(\overline{T}, D),$$
$$= P(v|T) - (P(v|T) - P(v|\overline{T}, D))P(\overline{T}, D).$$

Taking a closer look at $P(\overline{T}, D)$,

$$P(\overline{T}|D) = 1 - \frac{P(D|T)P(T)}{P(D|T)P(T) + P(D|\overline{T})P(\overline{T})},$$
$$\leq 1 - \frac{P(D|T)P(T)}{P(D|T)P(T) + \gamma^{-d}P(D|T)P(\overline{T})},$$
$$= 1 - \frac{\gamma^d P(T)}{\gamma^d P(T) + P(\overline{T})},$$
$$= \frac{P(\overline{T})}{\gamma^d P(T) + P(\overline{T})},$$
$$\approx \frac{P(\overline{T})}{P(T)} \gamma^{-d},$$

where this final approximation grows stronger as $d \to \infty$. Applying this bound to the above,

$$P(v|D) \geq P(v|T) - \gamma^{-d} \frac{(P(v|T) - P(v|\overline{T}, D))P(\overline{T})}{P(T)},$$

as desired. $\square$

### I.4 ACCELERATING HYPER-PARAMETER SEARCH

Now seeing some of the reasons why the scaling laws are of the form that fits well, we now want to show how it is possible to use these scaling laws to improve compute needs for the final hyper-parameter search. We start at a fairly strong assumption about the scaling laws, and leave the relaxation of this assumption (amounting to stronger bounds on the scaling laws from transfer learning literature) to future work.

**Assumption I.5.** The scaling law predictors $\hat{\pi}$ are accurate in the sense that for some fixed $\sigma^2$, $\pi(\theta) = \hat{\pi}(\theta) + \xi$ where $\xi \sim \mathcal{N}(0, \sigma^2)$.

This $\sigma^2$ will encapsulate many errors in the above, with the most notable being errors in predicting exact training performance of the model. Both $\sigma^2$ and $\hat{\pi}$ can be learned from sampling, assuming that the functional form of $\hat{\pi}$ is accurate.

Furthermore, let $s = \pi - \alpha C$ be the score function for a given model, which is random as a result of $\pi$ being random. While we can assume that over certain parameters, such as dataset size (or proportion) and training iterations, both $\pi$ and $C$ are monotonic, this no longer holds for $s$.

**Theorem I.6.** *Consider the situation where $s$ is being optimized using dataset proportion $\rho$ and number of iterations $I$. Assume access to $\hat{\pi}(\rho, I)$ and $C(\rho, I)$ and therefore $\hat{s}(\rho, I)$. Let $\mathcal{P}$ be some finite set of parameters to be searched over. Then, with probability $1 - \delta$, the optimal parameters for $s(\rho, I)$ will be found in the top $k$ of the $\hat{s}(\rho, I)$ when*

$$\hat{s}_{k+1} \leq \hat{s}_1 - 2\sigma \sqrt{\log \frac{2|\mathcal{P}|}{\delta}},$$

*where $\hat{s}_{k+1}$ is the $(k+1)$-largest and $\hat{s}_1$ is the largest of the $\hat{s}$.*

*Proof.* Let $\pi'(i)$ be the sorting permutation of $\{\hat{s}(\rho, I)|(\rho, I) \in \mathcal{P}\}$, where larger $\hat{s}$ occur at smaller $i$. For simplicity, denote $s_{\pi'(i)}$ as $s_i$ and $\hat{s}_{\pi'(i)}$ as $\hat{s}_i$.

Also, let $s^* = \max_i s_i$ with associated index $i^* = \arg\max_i s_i$. The goal is to show that the probability that $i^*$ is not in $[k]$ is small. Note that $s_1 - s_k \sim \mathcal{N}(\hat{s}_1 - \hat{s}_k, 2\sigma^2)$ which will be used later.

$$P(i^* \notin [k]) \leq \sum_{i=k+1}^{|\mathcal{P}|} P(s_i > s_j \text{ for } j \in [k])$$

$$\leq \sum_{i=k+1}^{|\mathcal{P}|} P(s_i > s_1)$$

$$\leq \sum_{i=k+1}^{|\mathcal{P}|} P(s_{k+1} > s_1)$$

$$\leq \sum_{i=k+1}^{|\mathcal{P}|} 2e^{-\frac{(\hat{s}_1 - \hat{s}_{k+1})^2}{4\sigma^2}}$$

$$\leq 2|\mathcal{P}|e^{-\frac{(\hat{s}_1 - \hat{s}_{k+1})^2}{4\sigma^2}}$$

For this to be bounded by $\delta$, the following need to hold.

$$2|\mathcal{P}|e^{-\frac{(\hat{s}_1 - \hat{s}_{k+1})^2}{4\sigma^2}} \leq \delta,$$

$$\frac{(\hat{s}_1 - \hat{s}_{k+1})^2}{4\sigma^2} \geq \log \frac{2|\mathcal{P}|}{\delta},$$

$$\hat{s}_{k+1} \leq \hat{s}_1 - 2\sigma \sqrt{\log \frac{2|\mathcal{P}|}{\delta}}.$$

$\square$

This shows we need tight control on the prediction variances to be sure we have attempted to train the model with the true optimal parameters.

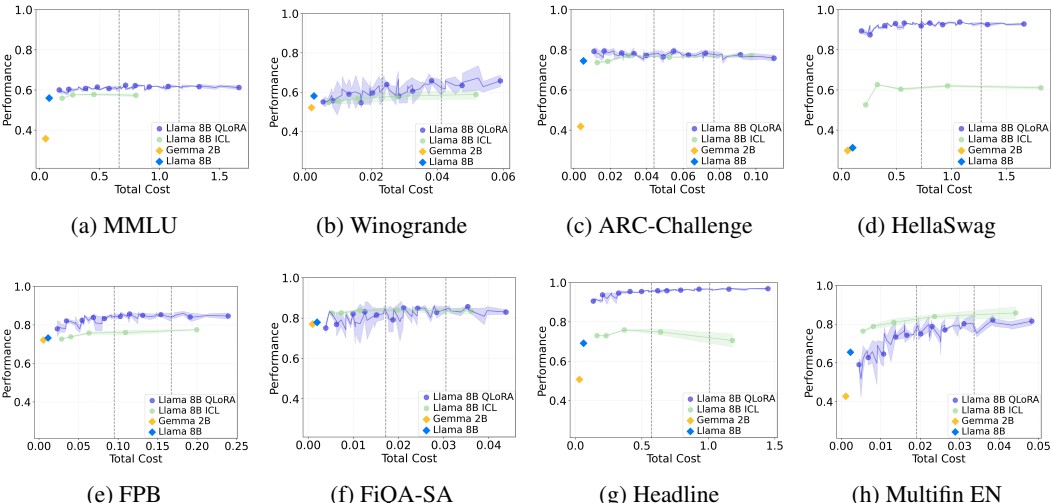

Figure 12: Actual QLoRA vs. ICL performance-cost trajectories across eight diverse tasks. Each plot presents the performance-cost curves for QLoRA (●) and ICL (●) on Llama 3 8B, with Gemma 2B (◆), and Llama 3 8B (◆) serving as baselines. Vertical dashed lines demarcate low, medium, and high-cost thresholds, determined by the minimum and maximum costs of both adaptation strategies. The shaded regions represent the standard deviation across 3 seeds for each configuration.

## J    FULL RESULTS FOR COMBINING TRAINING- AND TEST-TIME STRATEGIES

Following the four representative tasks shown in Section 5.4, we present in Figure 12 the full scaling behavior results across all eight tasks.

## K    EXTENDED POTENTIAL IMPLICATION

Fine-tuning large language models (LLMs) at scale presents significant computational and financial challenges. We analyze these costs through a practical case study of fine-tuning GPT-4o, using OpenAI's current pricing structure of $25 per million training tokens[3]. Following the training protocol established in the Llama 3 report (Dubey et al., 2024), a typical fine-tuning run requires an average of 8.75K steps with sequences of 8,192 tokens each, amounting to approximately 71M tokens per complete pass through the training data. To identify optimal fine-tuning parameters, practitioners typically need to explore various hyperparameter combinations. Consider a systematic exploration of epochs (ranging from 1 to 10) and data mixing strategies (10 options), resulting in 100 total trials. Each trial with a single epoch over the full dataset costs $1,775 in training compute alone, with costs scaling linearly with the number of epochs. Assume the validation phase requires processing queries totaling 1M tokens, with model outputs averaging 3x the input length due to multi-step reasoning. Given OpenAI's pricing of $2.5 per million input tokens and $10 per million output tokens, each validation run incurs an additional cost of $32.5. Across all trials, the validation cost sums to $3,250, while the total training cost reaches $976,250 (55 total epochs × 10 data mixing strategies × $1,775). This brings the total adaptation cost to approximately $979,500. Our method, COSMOS, substantially reduces these costs through accurate performance prediction using a small amount of data. In resource-intensive scenarios, our approach achieves a cost reduction of 95.95% compared to the total cost, enabling practitioners to estimate the performance-cost trade-offs of different configurations while only running a small subset of experiments. This reduces the total cost to approximately $39,670–a 24.7x reduction. This dramatic cost reduction enables practitioners to make informed decisions about the performance-cost trade-offs that best suit their specific requirements and constraints.

Table 8: COSMOS demonstrates robust performance across *275* model–strategy–configuration combinations, spanning diverse architectures (Gemma, Llama, Qwen, Mistral) and scales (2B-9B), maintaining high prediction accuracy (average MAE of 0.61% for HellaSwag and 1.16% for Multifin EN) while drastically reducing computational costs.

| Tasks | Cost Level | Pred. Acc (%) | Act. Acc (%) | MAE ↓ (%) | Act. Total Cost ($) | Ours Cost ($) | CRR ↑ (%) |
|---|---|---|---|---|---|---|---|
| | Low | 94.87 | 94.87 | 0.00 | 27.042 | 2.354 | 91.30 |
| HellaSwag | Medium | 94.71 | 95.64 | 0.93 | 62.838 | 2.470 | 96.07 |
| | High | 94.94 | 95.84 | 0.90 | 124.414 | 2.759 | 97.78 |
| | Low | 84.09 | 85.76 | 1.67 | 0.748 | 0.271 | 63.72 |
| Multifin EN | Medium | 85.91 | 86.36 | 0.45 | 1.751 | 0.265 | 84.88 |
| | High | 86.82 | 88.18 | 1.36 | 3.399 | 0.243 | 92.85 |

## L    EXPANDED MODEL EVALUATION: DIVERSE FAMILIES AND SCALES

To establish the generalizability of COSMOS, we significantly expand our evaluation framework beyond the initial Gemma 2B and Llama 3 8B models to encompass diverse model families and scales. This comprehensive evaluation now incorporates models from multiple architectures (Mistral Jiang et al. (2023) and Qwen Qwen (2024)), newer generations (Gemma 2 Gemma et al. (2024b)), and various parameter scales (7B to 9B). Following the experimental protocol established in Section 5, our expanded evaluation framework includes:

- Model pool: 5 models including Gemma 2B, Llama 3 8B, Gemma 2 9B, Qwen 2 7B, Mistral 7B v0.3
- Strategies: QLoRA and retrieval-augmented ICL
- QloRA configurations: Training iterations $\in \{4, 5, 6, 7, 8\}$ and data portions $\in \{0.1, 0.2, ..., 1.0\}$ at 0.1 increments
- ICL configurations: retrieved number of demonstrations $\in \{1, 2, 4, 8, 16\}$

This systematic design yields 275 model–strategy–configuration combinations (5 models × (50 QLoRA + 5 ICL)) per task, with all results averaged across three random seeds to ensure statistical robustness.

The results in Table 8 demonstrate COSMOS's consistent effectiveness across this expanded evaluation space even though we adopt the same training protocols for all models. For the general domain HellaSwag benchmark, COSMOS achieves an average Mean Absolute Error (MAE) of merely 0.61% across all three budget constraints, with perfect prediction accuracy (0% MAE) in the low-cost setting while reducing total evaluation costs by an average of 95.05%. Even for the domain-specific Multifin EN benchmark, which presents high variance scenarios (38-380 training samples), our method maintains strong performance with an MAE of 1.16%, demonstrating its robustness to data scarcity and domain shift.

## M    IS COSMOS ROBUST WITH LIMITED DATA ACCESS?

Table 9: COSMOS maintains high prediction accuracy with limited data: Using only 10% of training data achieves comparable MAE (0.24%) to full data access (0.20%) while further improving cost reduction (97.40% vs. 96.63%) across all budget constraints on HellaSwag.

| Task | Cost Level | Act. Acc (%) | Pred. Acc (%) (100% data) | MAE↓ (100% data) | Pred. Acc (%) (10% data) | MAE↓ (10% data) | Act. Cost ($) | Ours cost ($) (100% data) | CRR↑ (%) (100% data) | Ours cost ($) (10% data) | CRR↑ (%) (10% data) |
|---|---|---|---|---|---|---|---|---|---|---|---|
| | Low | 94.38 | 94.38 | 0.00 | 94.38 | 0.00 | 13.30 | 0.67 | 94.99 | 0.60 | 95.51 |
| HellaSwag | Medium | 94.11 | 93.68 | 0.43 | 94.11 | 0.00 | 17.28 | 0.52 | 96.99 | 0.30 | 98.25 |
| | High | 93.31 | 93.15 | 0.16 | 92.58 | 0.73 | 10.39 | 0.22 | 97.90 | 0.16 | 98.44 |
| Average | | | | 0.20 | | 0.24 | | | 96.63 | | 97.40 |

A practical question for deployment scenarios concerns whether COSMOS requires full access to the training dataset during performance prediction. While our main experiments utilized complete datasets for comprehensive evaluation, this design choice was motivated by convenience rather than

---

[3]https://openai.com/api/pricing/

Table 10: COSMOS demonstrates perfect prediction accuracy (0% MAE) on Multifin EN in both full and limited data scenarios, while the 10% data setting yields enhanced cost savings (84.71% vs. 83.91%), highlighting the method's robustness to data constraints in domain-specific tasks.

| Task | Cost Level | Act. Acc (%) | Pred. Acc (%) (100% data) | MAE↓ (100% data) | Pred. Acc (%) (10% data) | MAE↓ (10% data) | Act. Cost ($) | Ours cost ($) (100% data) | CRR↑ (%) (100% data) | Ours cost ($) (10% data) | CRR↑ (%) (10% data) |
|---|---|---|---|---|---|---|---|---|---|---|---|
| | Low | 80.91 | 80.91 | 0.00 | 80.91 | 0.00 | 0.287 | 0.083 | 70.92 | 0.082 | 71.41 |
| Multifin EN | Medium | 83.94 | 83.94 | 0.00 | 83.94 | 0.00 | 0.504 | 0.051 | 89.84 | 0.045 | 91.05 |
| | High | 85.76 | 85.76 | 0.00 | 85.76 | 0.00 | 0.360 | 0.033 | 90.95 | 0.030 | 91.67 |
| Average | | | | 0.00 | | 0.00 | | | 83.91 | | 84.71 |

necessity—our lightweight linear model for QLoRA trains efficiently regardless of data volume. Importantly, COSMOS's data portion is a configurable parameter that can be adjusted based on resource constraints in real-world applications.

To assess the data efficiency of our approach, we conduct a comparative analysis examining prediction performance when accessing 100% versus only 10% of the dataset during the prediction phase. Following our experimental protocol from Section 5, we evaluate three distinct cost constraints with a performance-prioritizing score function across a configuration space covering training iterations $\in$ {4, 5, 6, 7, 8} and data portions $\in$ {0.1, ..., 1.0}, with all results averaged over three random seeds for statistical robustness.

Tables 9 and 10 show that COSMOS's predictors are robust to limited data. For the general domain HellaSwag task, our method achieves an average MAE of just 0.24% across three budget levels when using only 10% of the data, with two perfect predictions (0% MAE). This approach further improved cost savings from 96.63% (with 100% data) to 97.40% (with 10% data). For the domain-specific Multifin EN challenge, our method perfectly predicts (0% MAE) the optimal strategies across all three cost levels while enhancing cost savings from 83.91% to 84.71%.

# N   ARE THERE SIGNALS TRANSFERRABLE BETWEEN MODELS?

To further test the generality of COSMOS, we investigate whether predictors trained on smaller models can be transferred to larger models within the same family. Specifically, we train predictors using results from Gemma 2B and then apply them directly to predict strategy performance for Gemma 2 9B, without collecting new embeddings or retraining predictors for the larger model. The only additional step is a lightweight calibration using a few small-scale runs (e.g., 10% of the training data).

**Intuition.**   Smaller and larger models from the same family often exhibit similar performance trends across adaptation strategies and configurations. If these signals transfer, it would allow practitioners to estimate large-model performance by reusing predictors trained on cheaper, small-scale runs.

**Results.**   Tables 11 and 12 compare results obtained with original predictors (trained directly on the target model) against transferred predictors (trained on Gemma 2B and calibrated for Gemma 2 9B). On both HellaSwag and Multifin EN, transferred predictors achieve virtually the same accuracy as the original setup, while further improving cost savings. For example, on HellaSwag the cost reduction ratio (CRR) improves from 96.02% to 96.38%, and on Multifin EN from 87.61% to 88.74%.

Table 11: Performance of COSMOS with original predictors trained directly on embeddings from Gemma 2 9B.

| Task | Cost Level | Pred. Acc. | Act. Acc. | MAE | Act. Cost ($) | Ours Cost ($) | CRR (%) |
|---|---|---|---|---|---|---|---|
| | Low | 95.11 | 95.64 | 0.53 | 9.857 | 0.615 | 93.76 |
| HellaSwag | Med | 95.39 | 95.60 | 0.21 | 21.449 | 0.724 | 96.63 |
| | High | 95.84 | 95.84 | 0.00 | 34.830 | 0.807 | 97.68 |
| | Low | 73.94 | 74.55 | 0.61 | 0.269 | 0.055 | 79.39 |
| Multifin EN | Med | 84.24 | 84.24 | 0.00 | 0.601 | 0.060 | 90.06 |
| | High | 84.55 | 85.45 | 0.90 | 0.974 | 0.064 | 93.39 |

Table 12: Performance of COSMOS using predictors transferred from Gemma 2B with lightweight calibration.

| Task | Cost Level | Pred. Acc. | Act. Acc. | MAE | Act. Cost ($) | Ours Cost ($) | CRR (%) |
|------|-----------|-----------|-----------|-----|--------------|--------------|---------|
| | Low | 95.11 | 95.64 | 0.53 | 9.857 | 0.559 | 94.33 |
| HellaSwag | Med | 95.39 | 95.60 | 0.21 | 21.449 | 0.658 | 96.93 |
| | High | 95.84 | 95.84 | 0.00 | 34.830 | 0.734 | 97.89 |
| | Low | 73.94 | 74.55 | 0.61 | 0.269 | 0.050 | 81.27 |
| Multifin EN | Med | 84.24 | 84.24 | 0.00 | 0.601 | 0.054 | 90.96 |
| | High | 84.55 | 85.45 | 0.90 | 0.974 | 0.059 | 93.99 |

**Takeaway.** These findings suggest that COSMOS can benefit from transferability across model scales, reducing the need to retrain predictors or rerun large-scale experiments for every model. This opens up a promising direction for scaling COSMOS to even larger models in a cost-effective manner, which we plan to explore further in future work.

## O   CAN COSMOS BE APPLIED TO OTHER ADAPTATION STRATEGIES?

We evaluate whether the two instantiated methods under COSMOS can be applied to other adaptation strategies beyond those used in the main paper. In this study, we apply LoRA (Hu et al., 2022) and dense retrieval for ICL with `all-MiniLM-L6-v2` on `Llama-3-8B-Instruct`, using the FPB benchmark.

**Setup.** We vary:

- Number of finetuning iterations: {4, 5, 6, 7, 8}
- Data portion: {0.1, 0.2, 0.3, 0.4, 0.5, 0.6, 0.7, 0.8, 0.9, 1.0}
- Number of ICL shots: {1, 2, 4, 8, 16}

This yields a total of 55 combinations.

**Results.** Table 13 shows the results. Across all cost regimes, COSMOS achieves strong prediction accuracy (MAE of 0.67% overall) while significantly reducing costs. In the high-cost setting, COSMOS achieves a $67.6\times$ reduction in computational expenditure.

Table 13: Results on FPB with Llama-3-8B-Instruct using LoRA and dense retrieval. COSMOS maintains high prediction accuracy (MAE 0.67%) while achieving substantial cost reductions (up to $67.6\times$).

| Task | Cost Level | Pred. Acc. | Act. Acc. | MAE | Act. Cost ($) | Ours Cost ($) | CRR (%) |
|------|-----------|-----------|-----------|-----|--------------|--------------|---------|
| | Low | 83.85 | 84.96 | 1.11 | 4.05 | 0.31 | 92.33 |
| FPB | Med | 85.34 | 86.01 | 0.67 | 12.32 | 0.31 | 97.49 |
| | High | 85.88 | 86.10 | 0.22 | 7.84 | 0.12 | 98.52 |

**Summary.** This experiment shows that the instantiated methods under COSMOS can be successfully applied to other adaptation strategies, such as LoRA and dense retriever, while retaining both predictive accuracy and efficiency.

## P   A CONCRETE EXAMPLE OF STRATEGY SELECTION BASED ON PREDICTED METRICS

In Tables 14, 15 and 16, we provide a concrete example illustrating how practitioners can select the optimal strategy based on COSMOS's predicted metrics. Using the HellaSwag dataset across three cost regimes with a performance-prioritizing function (Same setting in Section 5.1), we demonstrate

Table 14: COSMOS accurately predicts that QLoRA finetuning with 0.8 portion of data for 4 iterations yields optimal performance within the low cost budget. This strategy achieves the highest predicted accuracy (0.921) among all 30 available strategies, and validation confirms it indeed delivers the best actual performance (0.944). COSMOS predicts this strategy will cost $0.728, closely matching the actual cost of $0.725.

| Cost Level | Strategy | Data Portion | Iter | # shots | Pred. Acc | Act. Acc | Predicted Cost ($) | Act. Cost ($) |
|---|---|---|---|---|---|---|---|---|
| | | 0.1 | 4 | - | 0.894 | 0.894 | 0.181 | 0.181 |
| | | 0.1 | 5 | - | 0.890 | 0.890 | 0.201 | 0.200 |
| | | 0.1 | 6 | - | 0.885 | 0.885 | 0.221 | 0.220 |
| | | 0.1 | 7 | - | 0.882 | 0.882 | 0.240 | 0.239 |
| | | 0.1 | 8 | - | 0.875 | 0.875 | 0.260 | 0.259 |
| | | 0.2 | 4 | - | 0.898 | 0.922 | 0.259 | 0.258 |
| | | 0.2 | 5 | - | 0.893 | 0.907 | 0.298 | 0.297 |
| | | 0.2 | 6 | - | 0.888 | 0.909 | 0.337 | 0.336 |
| | | 0.2 | 7 | - | 0.885 | 0.910 | 0.376 | 0.375 |
| | | 0.2 | 8 | - | 0.878 | 0.907 | 0.415 | 0.413 |
| | | 0.3 | 4 | - | 0.905 | 0.923 | 0.337 | 0.335 |
| | | 0.3 | 5 | - | 0.900 | 0.921 | 0.395 | 0.393 |
| | | 0.3 | 6 | - | 0.895 | 0.916 | 0.454 | 0.452 |
| | QLoRA | 0.3 | 7 | - | 0.892 | 0.912 | 0.512 | 0.510 |
| Low | | 0.3 | 8 | - | 0.885 | 0.914 | 0.571 | 0.567 |
| | | 0.4 | 4 | - | 0.908 | 0.934 | 0.415 | 0.414 |
| | | 0.4 | 5 | - | 0.903 | 0.931 | 0.494 | 0.491 |
| | | 0.4 | 6 | - | 0.898 | 0.927 | 0.572 | 0.569 |
| | | 0.4 | 7 | - | 0.895 | 0.927 | 0.650 | 0.647 |
| | | 0.4 | 8 | - | 0.888 | 0.919 | 0.728 | 0.724 |
| | | 0.5 | 4 | - | 0.910 | 0.930 | 0.494 | 0.477 |
| | | 0.5 | 5 | - | 0.906 | 0.933 | 0.592 | 0.570 |
| | | 0.5 | 6 | - | 0.901 | 0.928 | 0.690 | 0.664 |
| | | 0.6 | 4 | - | 0.915 | 0.940 | 0.573 | 0.570 |
| | | 0.6 | 5 | - | 0.911 | 0.930 | 0.690 | 0.687 |
| | | 0.7 | 4 | - | 0.921 | 0.937 | 0.651 | 0.648 |
| | | **0.8** | **4** | **-** | **0.921** | **0.944** | **0.728** | **0.725** |
| | | - | - | 1 | 0.526 | 0.526 | 0.177 | 0.219 |
| | ICL | - | - | 2 | 0.591 | 0.627 | 0.262 | 0.326 |
| | | - | - | 4 | 0.617 | 0.604 | 0.432 | 0.538 |

Table 15: Predicted performance and cost given by COSMOS and actual performance and cost corresponding to each strategy within the medium cost level.

| Cost Level | Strategy | Data Portion | Iter | # shots | Pred. Acc | Act. Acc | Predicted Cost ($) | Act. Cost ($) |
|---|---|---|---|---|---|---|---|---|
| | | 0.5 | 7 | - | 0.898 | 0.926 | 0.787 | 0.758 |
| | | 0.5 | 8 | - | 0.891 | 0.919 | 0.885 | 0.851 |
| | | 0.6 | 6 | - | 0.906 | 0.933 | 0.807 | 0.803 |
| | | 0.6 | 7 | - | 0.903 | 0.926 | 0.925 | 0.920 |
| | | 0.6 | 8 | - | 0.896 | 0.922 | 1.042 | 1.037 |
| | | 0.7 | 5 | - | 0.917 | 0.933 | 0.788 | 0.784 |
| | | 0.7 | 6 | - | 0.911 | 0.928 | 0.924 | 0.920 |
| | | 0.7 | 7 | - | 0.908 | 0.929 | 1.061 | 1.056 |
| Medium | QLoRA | 0.7 | 8 | - | 0.901 | 0.926 | 1.198 | 1.192 |
| | | 0.8 | 5 | - | 0.917 | 0.936 | 0.884 | 0.880 |
| | | 0.8 | 6 | - | 0.912 | 0.934 | 1.041 | 1.036 |
| | | 0.8 | 7 | - | 0.909 | 0.930 | 1.197 | 1.191 |
| | | 0.9 | 4 | - | 0.925 | 0.941 | 0.807 | 0.803 |
| | | 0.9 | 5 | - | 0.921 | 0.936 | 0.983 | 0.978 |
| | | 0.9 | 6 | - | 0.915 | 0.929 | 1.158 | 1.153 |
| | | 1.0 | 4 | - | 0.931 | 0.937 | 0.886 | 0.881 |
| | | 1.0 | 5 | - | 0.928 | 0.939 | 1.081 | 1.076 |
| | ICL | - | - | 8 | 0.620 | 0.620 | 0.772 | 0.965 |

the decision-making process. In the low-cost regime (30 candidate strategies), our predicted values identify QLoRA with 0.8 data portion for 4 iterations as the optimal choice, and this is the actual optimal strategy. For medium-cost scenarios (18 candidates), COSMOS guides selection toward 1.0 data portion with 4 iterations (0.937 accuracy), which closely approximates the ground-truth optimal strategy of 0.9 data portion with 4 iterations (0.941 accuracy, MAE: 0.004). Similarly, in high-cost settings (7 candidates), our predicted best strategy (1.0 data portion, 6 iterations, 0.931 accuracy) nearly matches the optimal strategy (1.0 data portion, 7 iterations, 0.933 accuracy, MAE: 0.002). Note that while we present these examples with our performance-prioritizing objective, practitioners can define custom trade-off functions between performance and cost as described in Section 3.1.

Table 16: Predicted performance and cost given by COSMOS and actual performance and cost corresponding to each strategy within the high cost level.

| Cost Level | Strategy | Data Portion | Iter | # shots | Pred. Acc | Act. Acc | Predicted Cost ($) | Act. Cost ($) |
|---|---|---|---|---|---|---|---|---|
| High | QLoRA | 0.8 | 8 | - | 0.902 | 0.927 | 1.353 | 1.346 |
| | | 0.9 | 7 | - | 0.913 | 0.926 | 1.334 | 1.327 |
| | | 0.9 | 8 | - | 0.906 | 0.926 | 1.510 | 1.502 |
| | | 1.0 | 6 | - | 0.920 | 0.931 | 1.277 | 1.270 |
| | | 1.0 | 7 | - | 0.917 | 0.933 | 1.472 | 1.465 |
| | | 1.0 | 8 | - | 0.910 | 0.929 | 1.668 | 1.659 |
| | ICL | - | - | 16 | 0.620 | 0.611 | 1.452 | 1.815 |

## Q  EXAMPLES OF ADAPTATION STRATEGIES

**Training-time adaptation strategies.** A training-time adaptation strategy modifies the parameters of a language model $f_\theta$, where $\theta \in \Theta$. Examples include full fine-tuning to parameter-efficient methods like LoRA and QLoRA. Two techniques in Ex. 1 are both training-time methods. Formally, a training time adaptation function $T^{\text{tr}} : f_\theta \to f_{\theta'}$, transforms a base model into a task-specialized model.

**Test-time adaptation strategies.** Test-time (or inference time) adaptation strategies complement training-time approaches by modifying the input and/or output processing rather than the model parameters such as prompt tuning (Lester et al., 2021), CoT, and ICL. A test-time adaptation function $T^{\text{inf}} : \mathcal{X} \times \mathcal{Y} \to \mathcal{X}' \times \mathcal{Y}'$ transforms the input-output space to enhance model performance without parameter updates.

**Hybrid adaptation strategies.** Recent research demonstrates growing interest in hybrid adaptation strategies that fall into the intersection of both training-time and test-time adaptations (Soylu et al., 2024). Formally, hybrid approaches can be represented as composite adaptation functions where parameter transformations and input-output space modifications work in concert: $T^{\text{hybrid}} = T^{\text{tr}} \circ T^{\text{inf}}$.

**Model routing as a special case.** Model routing (*i.e.*, directing different queries to different models) can be viewed as a constrained instance of the strategy selection problem where: 1) The adaptation strategy pool contains a single strategy with fixed configuration; 2) The router operates at the query level rather than the task level.

## R  A DETAILED COMPARISON WITH RECENT SCALING-LAW BASED APPROACHES

Recent work has explored the use of scaling laws for model or configuration selection in fine-tuning and multimodal settings (Haowei et al., 2024; Zeng et al., 2025; He et al., 2025; Haowei et al., 2025). Here, we situate COSMOS within this emerging literature and highlight the key conceptual differences.

A first distinction concerns the *scope of the selection problem*. Methods such as Haowei et al. (2024) and Zeng et al. (2025) study model selection for SFT by predicting test loss as a function of data size via rectified scaling laws. He et al. (2025) focuses on selecting VLM components (vision encoder and LLM backbone), while Haowei et al. (2025) aims to automatically discover scaling laws that relate data size or configuration choices to loss. These works address selection within a single adaptation protocol, typically standard supervised fine-tuning.

COSMOS tackles a strictly broader problem. We consider a search space spanning model, adaptation strategy, and configuration jointly, covering training-time approaches (e.g., QLoRA, LoRA) and test-time approaches (e.g., retrieval-augmented ICL), along with their respective configuration knobs. From this perspective, the cited scaling-law methods can be seen as solving sub-instances of the COSMOS problem (e.g., model selection under fixed SFT). Indeed, their learned laws could be incorporated as strategy-specific predictors within COSMOS, illustrating complementarity rather than overlap.

A second difference lies in *cost awareness*. Existing scaling-law approaches are primarily designed to predict or compare performance (typically test loss). COSMOS explicitly models both downstream performance and end-to-end cost (adaptation, prediction, and evaluation), enabling selection under

user-defined performance–cost preferences. This makes the framework directly applicable in compute- or budget-constrained settings that the cited works do not address.

Third, the *predicted quantity* differs. Scaling-law methods predict test loss, and selection decisions are derived from this surrogate. One challenge that COSMOS tackles is the fact that the test loss is often not linearly correlated with downstream task performance (Liu et al., 2023; Ge et al., 2025). COSMOS instead predicts downstream performance directly, alongside cost, for each strategy–model–configuration tuple. This yields a more actionable signal for practitioners optimizing real task performance.

Finally, COSMOS operates under markedly different *data-efficiency assumptions*. Scaling-law approaches assume monotonic improvement of test loss with increasing data and typically fit their estimators using multiple observations across dataset fractions or training durations. COSMOS does not impose a monotonic relationship between data size and downstream performance for SFT and is designed for an extreme low-observation regime: in our experiments, a single real observation suffices for QLoRA prediction and two for retrieval-augmented ICL. Despite this minimal supervision, COSMOS achieves near-oracle selection accuracy (Appendix G). This makes COSMOS substantially more cost- and data-efficient, providing complementary advantages to methods that rely on richer traces of size–loss pairs.

