# OpenReview forum: "COSMOS: Are Performance–Cost Tradeoffs Predictable in Model--Strategy Selection?"
_ICLR.cc/2026/Conference — Submitted to ICLR 2026_

### Official Review · Reviewer_DXmw · 2025-10-18

**Soundness:** 3
**Presentation:** 2
**Contribution:** 3
**Rating:** 6
**Confidence:** 3

**Summary:**

This paper introduces COSMOS, a unified framework for predicting both performance and cost of LLM adaptation strategies without expensive exhaustive experimentation. The authors instantiate COSMOS with two complementary approaches: (1) embedding-augmented linear proxy models for QLoRA fine-tuning prediction, and (2) scaling law-based extrapolation for retrieval-augmented in-context learning. Experiments across 8 benchmarks demonstrate 92.72% average cost reduction with 1.09% mean absolute error in performance prediction.

**Strengths:**

1. The empirical study is extensive, covering diverse tasks (general reasoning and domain-specific financial tasks), multiple models (Gemma, Llama, Qwen, Mistral), and systematic configuration spaces.
2. The cost reduction (up to 98.71% in some scenarios) while maintaining prediction accuracy (MAE of 1.09%) is impressive and practically significant.
3. COSMOS provides a general abstraction that can accommodate different adaptation strategies and predictors, making it extensible.

**Weaknesses:**

1. While the paper claims to address a "multi-strategy" setting, the evaluation primarily focuses on only two strategies: QLoRA fine-tuning and retrieval-augmented ICL. How does COSMOS handle more complex strategies like Chain-of-Thought prompting variations, tree-search methods, or hybrid approaches mentioned in Section 2.
2. The paper makes specific architectural choices without sufficient justification: Why use bidirectional embeddings for causal language models in fine-tuning prediction? For ICL, why assume exponential saturation specifically? Have other functional forms been explored?
3. While Appendix G provides comparisons with search-based methods and scaling-law predictors, more recent methods could be included.

**Questions:**

1. How does COSMOS handle newly released models not in the training set for predictors?
2. Can the framework predict when fine-tuning might hurt performance (negative transfer)?
3. What is the computational cost of the prediction framework itself at scale?
4. How do predictions degrade when task distribution differs significantly from training data?
5. The work is limited to models in the 2B-9B parameter range. Could you discuss whether the outperformance of COSMOS holds for frontier-scale models (llama 13B, QwQ-32B)?
6. Two highly relevant recent works should be included: (a) LENSLLM: Unveiling Fine-Tuning Dynamics for LLM Selection and (b) EvoSLD: Automated Neural Scaling Law Discovery With Large Language Models.

---

> ### Author Response · Authors · 2025-11-28
> **Response to reviewer DXmw (1/4)**
>
> Thank you for your time and constructive feedback and for recognizing our extensive study, impressive and practically significant results, and extensiblity of COSMOS! We have addressed your concerns point by point below:
>
> > W1: How does COSMOS handle more complex strategies?
>
> Thank you for raising this important point! To directly address the reviewer's concern, we extend our evaluation to **open-ended mathematical reasoning (AIME2024)** using a **larger model** (s1.1-32B, a fine-tuned Qwen2.5-32B-Instruct) and **two new adaptation strategies**:
> * **s1**: force the model to continue reasoning by injecting “Wait” until the generation-budget limit is reached [1],
> * **Self-consistency**: generate multiple reasoning traces and aggregate them under a total budget [2].
>
> We proposed and instantiate COSMOS with lightweight **scaling-law based predictors**, using **minimal real observations**:
>
> * For **s1**: $\text{Acc} = a \cdot \log(\text{budget}+1) + b$, fit with *two* points (budget = 1k, 8k).
> * For **self-consistency**: $\text{Acc} = \big[a_0 \log(k+1)+a_1\big] \cdot \log(\text{budget}+1) +
>   \big[b_0 \log(k+1)+b_1\big],$
>   fit with *four* points (k ∈ {2,5}, budgets ∈ {1k, 8k}).
>
> We evaluate a broad strategy space:
> - s1 budgets: {1k, 2k, 4k, 8k, 16k, 30k, 32k}
> - Self-consistency sample counts: {1,2,...,20,30,50,60,100,150,200,300} and per sample budget: {1k, 2k, 4k, 8k, 16k, 30k, 32k}
> - Total-token constraint per sample: ≤ 320k tokens
>
> This yields **134 strategy candidates**. Costs are measured in tokens (actual + estimated), and we use the same score function (l.374) as in the main paper. Besides single round selection as we used in our main paper, we also show how COSMOS can seamlessly combine with search-based methods and conduct multi-round selection (up to 4 rounds; stop when two consecutive rounds yield no improvement) using greedy search.
>
> COSMOS instantiations achieve **perfect strategy selection** across all 11 budget levels with **MAE = 0.0%** at every budget level and cost reduction up to **30.3×**.
>
> | Task     | Budget Level (tok/sample) | Pred. Acc | Act. Acc | MAE$\downarrow$ | Act. Total Cost (k tok) | Ours Cost (k tok) | CRR $\uparrow$ (%) |
> | -------- | ------------------ | --------- | -------- | --------------- | ----------------------- | ----------------- | ------------------ |
> | AIME2024 | 2k                 | 6.0%      | 6.0%     | 0.0%            | 150                     | 48                | 67.8%              |
> |          | 5k                 | 15.0%     | 15.0%    | 0.0%            | 750                     | 129               | 82.8%              |
> |          | 8k                 | 30.0%     | 30.0%    | 0.0%            | 2280                    | 243               | 89.4%              |
> |          | 10k                | 30.0%     | 30.0%    | 0.0%            | 3150                    | 291               | 90.8%              |
> |          | 16k                | 37.7%     | 37.7%    | 0.0%            | 8640                    | 501               | 94.2%              |
> |          | 20k                | 37.7%     | 37.7%    | 0.0%            | 12600                   | 612               | 95.1%              |
> |          | 32k                | 40.0%     | 40.0%    | 0.0%            | 25380                   | 855               | 96.6%              |
> |          | 100k (Multi-round) | 50.0%     | 50.0%    | 0.0%            | 83406                   | 5274              | 91.3%              |
> |          | 200k (Multi-round) | 56.7%     | 56.7%    | 0.0%            | 183261                  | 12999             | 91.0%              |
> |          | 300k (Multi-round) | 56.7%     | 56.7%    | 0.0%            | 302448                  | 11112             | 96.3%              |
> |          | 320k (Multi-round) | 60.0%     | 60.0%    | 0.0%            | 327300                  | 29400             | 89.4%              |
>
> [1] https://arxiv.org/abs/2501.19393
>
> [2] https://arxiv.org/abs/2203.11171

---

> ### Author Response · Authors · 2025-11-28
> **Response to reviewer DXmw (2/4)**
>
> > W2: Why use bidirectional embeddings for causal language models in fine-tuning prediction? For ICL, why assume exponential saturation specifically? Have other functional forms been explored?
>
> Thank you for this insightful feedback! These choices were made based on practical considerations:
> - Bidirectional embeddings: We use bidirectional attention because it provides richer contextual representations for each data entry, which improves the quality of the base-performance estimate used in the QLoRA predictor.
> - Exponential saturation for ICL: In our experiments (5 models × 8 benchmarks × 5 configs × 3 seeds), retrieval-augmented ICL consistently shows diminishing returns as shot count increases. We tested several other functional forms (log, power law, log–log, linear, quadratic) and found that an exponential-saturation curve fits this empirical trend best.
>
> > W3&Q6: While Appendix G provides comparisons with search-based methods and scaling-law predictors, more recent methods could be included. Two highly relevant recent works should be included: (a) LENSLLM: Unveiling Fine-Tuning Dynamics for LLM Selection [3] and (b) EvoSLD: Automated Neural Scaling Law Discovery With Large Language Models [4].
>
> These are great related work, and we appreciate the suggestions! **We have included them in the revised related work (Section 2) and uploaded a new version of our manuscript!** Below, we clarify how **COSMOS and these methods are complementary**:
> - **Scope of the selection problem (model-only vs. model–strategy–configuration)**: [3] focus primarily on model selection for fine-tuning, using rectified scaling laws to predict test loss as a function of SFT data size. COSMOS also performs model selection, but over a **strictly larger space**: it jointly selects over **model × strategy × configuration**, including training-time strategies (e.g., QLoRA, LoRA), test-time strategies (e.g., ICL variants), and rich configuration knobs (iterations, data portion, LoRA rank, shots, budgets, etc.). In this sense, [3] can be viewed as tackling a subset of the COSMOS selection space (model under a fixed adaptation protocol), and **these predictors can be plugged in as one particular strategy-specific instantiation inside COSMOS**. [4] aims to discover scaling laws (e.g., data size, configs -> loss) using LLMs. These laws could also serve as performance predictors inside COSMOS for a fixed strategy, though the selection problem evaluated in [4] primarily concerns model/configuration selection under a single training protocol, similar to [3].
> - **Cost-awareness as a first-class objective**: Both [3] and [4] predict or compare performance only. COSMOS is explicitly cost-aware: the framework models both downstream performance and end-to-end adaptation cost (adaptation, evaluation, prediction) and selects according to user preferences. This dimension is not addressed in [3,4].
> - **What is predicted: test loss vs. direct downstream performance & cost**: [3] and [4] use variants of rectified scaling laws to predict test loss of SFT. One challenge that COSMOS tackles is the fact that *the test loss is often not linearly correlated with downstream task performance* [5-6]. COSMOS instead directly predicts downstream performance together with cost for each strategy–model–configuration tuple. This provides a more direct signal for users who care about task-level metrics (e.g., accuracy) and allows flexible scoring based on their own performance–cost preferences.
> - **Assumptions and data efficiency**: The scaling-law approaches in [3-4] assume a monotonic relationship in test loss as data size increases, and their estimators are fitted using multiple observations across different dataset fractions or training schedules. COSMOS’s fine-tuning instantiation does **not** impose a monotonic relationship between data size and downstream performance, which allows it to capture more irregular behaviors observed in practice. Moreover, COSMOS is designed for an extreme low-observation regime: in our experiments, we use only one real observation for QLoRA and two observations for ICL to fit the predictors, yet still obtain near-oracle selection (Section 5, Appendix G). This makes COSMOS instantiations substantially more data- and cost-efficient, and complementary to methods that rely on richer size–loss traces.
>
>
> [3] https://arxiv.org/abs/2505.03793
>
> [4] https://arxiv.org/abs/2507.21184
>
> [5] https://arxiv.org/abs/2210.14199
>
> [6] https://arxiv.org/abs/2506.13216

---

> ### Author Response · Authors · 2025-11-28
> **Response to reviewer DXmw (3/4)**
>
> > Q1: How does COSMOS handle newly released models not in the training set for predictors?
>
> Thank you for the question! If the reviewer is asking whether COSMOS can make predictions for a completely unseen model with zero observations, this is an interesting direction that remains an open challenge: existing scaling-law approaches [3,4], and training-based methods [7] all require at least some observation for any new model.
>
> However, we have studied whether there are some signals transferable between models in Appendix N. We train predictors on Gemma-2B and apply them directly to Gemma-2 9B without collecting new embeddings or retraining for Gemma-2 9B.
> Results are promising. On both HellaSwag and Multifin EN, transferred predictors achieve the **same accuracy** as the original setup, while further improving cost savings. For example, on HellaSwag the cost reduction ratio (CRR) improves **from 96.02% to 96.38%**, and on Multifin EN **from 87.61% to 88.74%**.
>
> This suggests that models within the same family can potentially share similar performance trends across strategies and configurations, enabling effective transfer from small, inexpensive runs to larger models.
>
> Building on these ideas to extend COSMOS to fully zero-observation model prediction is an interesting direction for future work.
>
> [7] https://arxiv.org/abs/2406.09334
>
>
> > Q2: Can the framework predict when fine-tuning might hurt performance (negative transfer)?
>
> Yes. COSMOS instantiations can capture cases where fine-tuning hurts performance. For example, in ARC-Challenge (Figure 6), the fine-tuning curve decreases as compute increases, and the predictor tracks this downward trend. Winogrande and Multifin EN also show sharp non-monotonic fluctuations when scaling compute, and our predictor closely follows these variations as well. This demonstrates that the framework does not assume monotonic improvement and can model negative transfer effectively.
>
>
> > Q3: What is the computational cost of the prediction framework itself at scale?
>
> Thank you for this insightful question! The COSMOS framework itself does not prescribe a specific cost model (this choice is flexible and can be made by the practitioner), but for the two instantiations used in the main paper, the prediction cost is dominated by obtaining the real observations (i.e., the adaptation and evaluation runs). Thus, the total prediction cost scales linearly with the number of observations collected. Training the lightweight predictor (e.g., a linear layer for QLoRA or a small curve fit for AUG-ICL) is pretty cheap, typically seconds to minutes, and does not contribute significantly to the overall cost.
>
> > Q4: How do predictions degrade when task distribution differs significantly from training data?
>
> Thank you for this insightful question! In general, prediction-based methods, including training-based approaches [7] and scaling-law methods [3,4], assume that the observations used for fitting come from the same or a similar distribution as the configurations they are used to predict. COSMOS follows the same standard assumption; distribution shift is orthogonal to our contribution and is a shared limitation of existing predictive approaches. Addressing strong distribution shifts is an interesting direction for future work, but not the focus of this paper.

---

> ### Author Response · Authors · 2025-11-28
> **Response to reviewer DXmw (4/4)**
>
> > Q5: The work is limited to models in the 2B-9B parameter range. Could you discuss whether the performance of COSMOS holds for frontier-scale models (llama 13B, QwQ-32B)?
>
> Thank you for this question! Llama-3-8B-Instruct is already a more capable model than Llama-2-13B on standard benchmarks [8], so our evaluation includes models that are competitive with or stronger than the 13B class.
>
> As stated in l.820-831, validating COSMOS involved a very large number of experiments, which would be *prohibitive (in terms of compute and storage)* since *evaluating each configuration requires full training and inference*, E.g., across 5 models, 8 benchmarks, 3 seeds, 10 data portions, 5 iterations, and 5 LoRA r values, we tuned approximately 7,100 checkpoints, and our current study has already consumed over **10,000 GPU hours**. Extending this grid to larger models is currently infeasible within reasonable compute budgets.
>
> We **also evaluated a 32B model (s1.1-32B) in response to W1, and COSMOS instantiations achieved perfect strategy selection** (MAE = 0.0%) across 11 budget levels with up to 30.3× cost reduction, suggesting good scalability.
>
> We also note that COSMOS can naturally support multi-round or search-based selection, if prediction accuracy degrades, practitioners can simply adjust the fitted predictor (e.g., refit the anchor point or re-estimate parameters with the new observations from previous rounds).
>
> We will open-source all checkpoints and raw results to support further exploration at larger scales. In addition, our study already covers more modern and larger LLMs than prior predictive-selection work (e.g., [3][4] evaluate models up to 6.7B).
>
> [8] https://huggingface.co/meta-llama/Meta-Llama-3-8B-Instruct

---

### Official Review · Reviewer_nNYi · 2025-10-26

**Soundness:** 2
**Presentation:** 3
**Contribution:** 2
**Rating:** 4
**Confidence:** 3

**Summary:**

The paper introduces COSMOS, a unified framework that aims to predict the performance and cost trade-offs of various model and adaptation strategy combinations when deploying large language models (LLMs). Given the complexity of selecting models and adaptation strategies under resource constraints, the paper proposes a system to efficiently predict both performance and cost, thereby reducing the need for expensive trial-and-error experimentation. COSMOS is evaluated through two major adaptation strategies: QLoRA fine-tuning and retrieval-augmented in-context learning (ICL). The results show that COSMOS can accurately predict performance outcomes with high accuracy while reducing computational costs by up to 98.7%.

**Strengths:**

The paper introduces an interesting approach to reduce the computational costs of model and strategy selection by leveraging performance prediction. This is valuable since running full-scale experiments for every model and strategy combination can be prohibitively expensive. The idea of predicting performance and cost trade-offs without the need for exhaustive testing can save substantial resources. Additionally, the framework is clearly presented, and the experiments, though limited, demonstrate that the COSMOS method is able to predict performance effectively in the cases of QLoRA and ICL. For simpler tasks, the method shows promise in delivering high-quality predictions efficiently.

**Weaknesses:**

One significant weakness of the proposed framework is the over-simplification of the predictive models used for performance estimation. In the case of QLoRA fine-tuning, the use of a linear proxy model to predict performance may not be sufficiently accurate, especially for more complex tasks. The relationship between adaptation strategies (like QLoRA) and performance is often non-linear, involving interactions between hyperparameters such as learning rate and number of training steps. A linear model, despite its computational efficiency, fails to capture these intricate dynamics, leading to potential inaccuracies in performance predictions.

Similarly, for retrieval-augmented in-context learning (ICL), the scaling law-based prediction method relies on the assumption that performance improves linearly with the number of examples (shots). However, this assumption may not hold true in many real-world tasks. The effectiveness of ICL often depends on factors such as the quality of retrieved examples, the complexity of the task, and the nature of the query context. These factors are not adequately captured by a simple scaling law, which could lead to overestimations or underestimations of ICL's performance, particularly in complex scenarios.

Moreover, the experimental validation is primarily focused on QLoRA and ICL, which severely limits the generalizability of the framework. The framework's performance on other adaptation strategies or more diverse tasks remains unexplored. Without testing on a wider variety of models, strategies, and tasks, it is difficult to assess the framework's robustness and scalability. This narrow experimental scope reduces confidence in the framework's ability to generalize across different settings.

Additionally, both methods rely on relatively small-scale data for calibration and validation. Given that the predictions are based on proxy models with limited training and validation sets, there is a significant risk of overfitting. If the models are trained on a narrow range of examples, they may not generalize well to unseen tasks or larger models, further undermining the reliability of predictions.

**Questions:**

The experiments focus primarily on QLoRA and ICL. How well does COSMOS generalize to other adaptation strategies, have you tested the framework on a broader range of adaptation strategies and tasks?
Given the reliance on a linear proxy model, how accurate are the performance predictions when applied to tasks with high complexity or large models? Are there any plans to explore more advanced prediction methods to improve the accuracy of predictions?
How does COSMOS handle more complex adaptation strategies or tasks such as those seen in large-scale model fine-tuning? Can the framework be extended to handle these more complex relationships?

---

> ### Author Response · Authors · 2025-11-28
> **Response to Reviewer nNYi (1/3)**
>
> Thank you for your time and feedback and for recognizing that COSMOS is an interesting approach, is valuable, that our presentation is clear and we have promising results! We have addressed your concerns point by point below:
>
> > W1&Q2: Concern about over-simplification of the QLoRA performance predictor. Given the reliance on a linear proxy model, how accurate are the performance predictions when applied to tasks with high complexity or large models?
>
> Thank you for raising this important point! We would like to clarify that although the final predictor includes a linear layer, the **predicted performance is not linear**, nor do we train a single global linear model across all tasks, models, or configurations.
>
> **How the predictor works** (l.255-269):
> - For each (task, model, setup), we first obtain contextual sequence embeddings from the transformed model. These embeddings already capture rich, nonlinear behavior from the underlying language model.
> - A lightweight predictor is then trained locally on the output/answer space to obtain a base performance estimate. We do not train across hyperparameters or pool data across configurations.
> - A single shared pilot run is used for affine calibration, producing the final performance estimate used for model–strategy selection.
>
> Because each predictor is trained independently per setup, it can track non-linear performance variations across hyperparameters without assuming linear relationships in QLoRA dynamics. Empirically, Figure 6 shows that predicted performance closely follows non-monotonic actual behavior as the data and compute scale.
>
> The predictor is intentionally lightweight to remain extremely cost-efficient—we assume only one real fine-tuning run is available. More complex non-linear predictors would overfit under this minimal-observation regime and violate the efficiency goals of COSMOS.
>
> Despite using such minimal supervision, COSMOS instantiations still achieve **near-oracle selection accuracy (99.3%–100%)** and reduce computational cost by up to **49.1x** compared to eight baselines (Appendix G).
>
> > W2: Concern about the simplicity of the ICL scaling-law predictor
>
> Thank you for raising this thoughtful point! We would like to clarify that the functional form we use for ICL is **not arbitrary**, and it is based on consistent empirical patterns observed in our experiments (Appendix H.3). Across 5 models, 8 benchmarks, 5 configurations, and 3 seeds (over 600 combinations), retrieval-augmented ICL performance exhibits a clear diminishing-returns trend: large gains occur at small shot counts, followed by progressively smaller improvements as more examples are added. The exponential-saturation curve we use provides a stable and accurate fit to this widely observed behavior.
>
> It is also worth noting that this type of assumption is standard across scaling-law based approaches. Prior work similarly adopts specific functional forms to model how test loss scales with dataset size or model size[1,2]. Like those methods, we select a form that captures the dominant empirical trend while remaining data-efficient and easy to fit.
>
> However, our goal in COSMOS is slightly different from classical scaling-law studies: **We aim to make effective strategy selections when only a tiny handful of real observations are available**.
>
> Under this strict budget, complex predictors would overfit, while simple, empirically supported forms tend to generalize better. The exponential-saturation form allows us to extrapolate reliably from minimal data and is sufficient for comparing strategies at the level required for COSMOS.
>
> Importantly, COSMOS performs model–strategy selection, not fine-grained optimization within a single ICL strategy. Even a lightweight, data-efficient predictor can provide the **correct relative ranking**, and empirically, COSMOS achieves **near-oracle selection accuracy** under this regime while drastically reducing cost by up to 49.1x (Appendix G).
>
> The framework itself is **flexible**: if more real observations are available, users can adopt richer functional forms or iterative refinement. COSMOS only requires predictors for performance and cost; the specific modeling choice is not fixed.
>
> We will clarify these points and the connection to standard scaling-law assumptions in the revised manuscript.
>
> [1] https://arxiv.org/abs/2001.08361
>
> [2] https://arxiv.org/abs/2402.02314

---

> ### Author Response · Authors · 2025-11-28
> **Response to Reviewer nNYi (2/3)**
>
> > W3&Q1&Q4: Moreover, the experimental validation is primarily focused on QLoRA and ICL, which severely limits the generalizability of the framework. The framework's performance on other adaptation strategies or more diverse tasks remains unexplored. Without testing on a wider variety of models, strategies, and tasks, it is difficult to assess the framework's robustness and scalability. This narrow experimental scope reduces confidence in the framework's ability to generalize across different settings. The experiments focus primarily on QLoRA and ICL. How well does COSMOS generalize to other adaptation strategies, have you tested the framework on a broader range of adaptation strategies and tasks? How does COSMOS handle more complex adaptation strategies or tasks such as those seen in large-scale model fine-tuning? Can the framework be extended to handle these more complex relationships?
>
> Thank you for this raising this important point! We would first like to clarify the distinction between the **COSMOS framework** and the **specific predictors instantiated under it**.
>
> As stated in the remark at the beginning of Section 5 (l.295-302), COSMOS, as introduced in Section 3, is an **analysis framework/abstraction for approaching the model–strategy selection problem through predictive modeling**. The framework itself makes *no assumptions* about which adaptation strategies are used or how the corresponding performance and cost predictors must be designed (l.174-l.189). It formalizes how *any* compatible predictors for performance and cost can be combined to support flexible strategy selection under user-specified tradeoffs.
>
> In Section 4, we then **propose two concrete instantiations** of COSMOS for two representative and widely used adaptation strategies, QLoRA and retrieval augmented ICL. These are intended as **examples** of how COSMOS can be instantiated, not as the framework itself. The purpose is to examine *whether efficient, extremely low-observation predictors exist for prominent training-time and test-time adaptation strategies* (l.240-241), and Section 5 shows that this is indeed feasible. This demonstrates that it is indeed possible to approach the model–strategy selection problem via predictive modeling. Importantly, we do not claim that the two predictors will generalize to all possible adaptation strategies; rather, our contribution is a flexible and extensible formulation plus two concrete instantiations demonstrating feasibility.
>
> To directly address the reviewer’s concern, we also evaluate COSMOS on **other strategies, larger models, and new tasks**:
> 1. **LoRA** and a **dense retriever** (Appendix O):
>    COSMOS achieves high accuracy (MAE **0.67%**) while reducing cost by up to **67.6×**, indicating that the framework instantiations readily extend to similar adaptation methods.
>
> | Task | Cost Level | Pred. Acc | Act. Acc | MAE$\downarrow$ | Act. Total Cost ($) | Ours Cost ($)|  CRR $\uparrow$ (%) |
> |------|------------|----------|---------|----------|------------|-----------------|-------|
> | FPB | Low | 83.85% | 84.96% | 1.11% | 4.05  | 0.31 |  92.33% |
> | | Med | 85.34% | 86.01% | 0.67% |  12.32 | 0.31  | 97.49% |
> | | High | 85.88% | 86.10% | 0.22% | 7.84 | 0.12 | 98.52% |
>
> 2. **Open-ended generation and a larger model (AIME2024, 32B model)**:
>    We further instantiate COSMOS for open-ended mathematical reasoning on AIME2024 using a fine-tuned Qwen2.5-32B (s1.1-32B) and two new strategies:
>     - **s1** (forced continuation) [3]
>     - **Self-consistency** [4]
>
> Using only 2–4 real observations, we evaluate **134 strategy candidates** and obtain perfect strategy selection (MAE = **0.0%**) with up to **30.3×** cost reduction.
>
> |Task|Budget Level (tok/sample)|Pred. Acc|Act. Acc|MAE$\downarrow$| Act. Total Cost (k tok)|Ours Cost (k tok)|CRR $\uparrow$ (%)|
> |-|-|-|-|-|-|-|-|
> |AIME2024|2k| 6.0% | 6.0%|0.0%|150|48|67.8%|
> || 5k| 15.0% |15.0%|0.0%|750| 129| 82.8%|
> || 8k| 30.0%| 30.0% | 0.0%| 2280| 243| 89.4% |
> || 10k| 30.0%| 30.0%| 0.0%| 3150| 291| 90.8%|
> || 16k| 37.7% | 37.7%| 0.0% | 8640| 501| 94.2%|
> || 20k | 37.7%| 37.7%| 0.0% | 12600| 612| 95.1%|
> || 32k| 40.0%| 40.0%| 0.0%| 25380| 855| 96.6% |
> || 100k (Multi-round) | 50.0% | 50.0% | 0.0%|83406| 5274| 91.3%|
> || 200k (Multi-round) | 56.7%| 56.7%| 0.0%| 183261|12999|91.0%|
> || 300k (Multi-round) | 56.7%| 56.7%| 0.0%| 302448| 11112| 96.3%|
> || 320k (Multi-round) | 60.0%| 60.0%|0.0%|327300| 29400| 89.4%|
>
> Together, these additional experiments show that COSMOS extends naturally beyond QLoRA and ICL to **open-ended generation, multi-round selection, larger models, and more complex reasoning strategies**, while remaining extremely cost-efficient.
>
> [3] https://arxiv.org/abs/2501.19393
>
> [4] https://arxiv.org/abs/2203.11171

---

> ### Author Response · Authors · 2025-11-28
> **Response to Reviewer nNYi (3/3)**
>
> > W4: Concern about overfitting due to small calibration/validation sets
>
> Thank you for raising this concern! We believe there may be a misunderstanding about how predictors are trained and applied within COSMOS, and we are happy to clarify!
>
> Our approach **does not train a global predictor or navigator** over (model, configuration, performance) observations, nor do we build a single model that must generalize across heterogeneous tasks or strategy families (as in [5]). Instead, **each predictor is trained locally and minimally** for the specific adaptation strategy and setup under consideration (l.174-l.189).
> * For QLoRA, the predictor is trained on representations extracted *for that particular (task, model, setup)* and is then aligned using one shared pilot run.
> * For ICL, the predictor is fit using only 1–2 real observations for the strategy being estimated.
>
> Because we **do not learn a universal mapping** over the entire search space, the form of overfitting described in the comment, where a single selector is trained on a narrow range of examples and fails to generalize, does not arise. Each predictor is only used for local estimation within its own strategy, not across the full search space.
>
> [5] https://arxiv.org/abs/2406.09334
>
> > Q3: Are there any plans to explore more advanced prediction methods to improve the accuracy of predictions?
>
> Thank you for the question! We clarify that the two predictors used in the paper are **instantiations of COSMOS**, not the framework itself. COSMOS is a general predictive modeling abstraction for model–strategy selection: *any* method that predicts performance and cost can be integrated.
>
> In this work, we focus on the **extreme low-observation regime**. To study this setting meaningfully, we intentionally use simple, robust predictors that generalize reliably under such sparsity. Using highly expressive models would overfit with so little supervision and would not help answer our central question: *can effective strategy selection be achieved with minimal data?*
>
> Despite this minimal supervision, the COSMOS instantiations already achieve **near-oracle selection accuracy (99.3%–100%)** and up to **49.1× compute savings** (Appendix G), outperforming eight search-, training-, and scaling-law baselines.
>
> COSMOS itself is **fully compatible with more advanced predictors**, and exploring richer designs is a natural direction for future work.
>
> We hope these address your concerns, and we are happy to answer any further questions you may have!

---

### Official Review · Reviewer_MzTr · 2025-10-27

**Soundness:** 3
**Presentation:** 2
**Contribution:** 2
**Rating:** 2
**Confidence:** 3

**Summary:**

This paper addresses the "strategy selection problem": the challenge of choosing the optimal combination of a large language model and an adaptation strategy (e.g., fine-tuning, in-context learning) to balance performance and cost for a specific task. Exhaustively evaluating all combinations is computationally prohibitive. The authors propose COSMOS, a unified prediction framework designed to efficiently estimate the performance and cost of various model-strategy pairs at a minimal cost. The paper instantiates and studies this framework in two popular scenarios: 1) predicting QLORA fine-tuning performance using an embedding-augmented lightweight proxy model (akin to linear probing), and 2) predicting retrieval-augmented in-context learning (ICL) performance using low-sample scaling laws (an exponential saturation curve).

**Strengths:**

The paper effectively formalizes the "strategy selection problem", which is a practical and important challenge for researchers and practitioners.

**Weaknesses:**

The core technical contribution of the paper feels limited in its originality. The two prediction methods instantiated within the COSMOS framework are essentially linear probing on frozen embeddings and fitting a simple exponential scaling law. These are both well-established, non-novel techniques. The contribution is thus an *application* of these methods rather than the development of a new prediction framework.

The choice of experimental scenarios significantly limits the practical impact of the findings. The paper focuses on Q-LoRA and retrieval-augmented ICL, which are already low-cost adaptation strategies. The real-world "strategy selection problem" is most painful when resource-intensive options like full fine-tuning are on the table. The paper motivates this with a GPT-4o case study in Appendix K but provides no empirical validation for this high-stakes, more practical scenario. It is unclear if the linear proxy predictor, which relies on *frozen* embeddings, would be at all predictive for full fine-tuning, where the embeddings themselves are updated.

The formalization of the cost analysis framework in Section 3.3 is overly complex and not well-motivated. By trying to create a unified model based on fluctuating real-world prices, the framework lacks principled stability. A more robust analysis based on standardized computational units, such as FLOPs, would be more generalizable and principled, even if it's harder to map to a direct dollar amount.

The related work section on scaling laws and performance prediction is incomplete. It overlooks several recent and highly relevant papers that also focus on general-purpose performance prediction for model selection using efficient trials (e.g., [1,2,3]). A discussion of these works is necessary to properly contextualize the novelty and contribution of COSMOS.

[1] [Selecting large language model to fine-tune via rectified scaling law](https://arxiv.org/abs/2402.02314)

[2] [Mordal: Automated Pretrained Model Selection for Vision Language Models](https://arxiv.org/abs/2502.00241)

[3] [LENSLLM: Unveiling Fine-Tuning Dynamics for LLM Selection](https://arxiv.org/abs/2505.03793)

**Questions:**

1. Could you provide a simpler, more intuitive explanation of the cost framework? Why was this complex, price-based model chosen over a more standardized and stable one like total FLOPs? How do you account for the high volatility of the prices you use (e.g., GPU spot markets on Vast.ai) in a generalizable framework?
2. Your key insight for ICL prediction is that its performance "typically follows an exponential saturation curve". What is the empirical or theoretical basis for this specific functional form? This is a foundational assumption for half of your paper's experimental validation, and it is presented without supporting evidence or citation.
3. Your paper is motivated by the high cost of strategies like full fine-tuning (as shown in Appendix K), yet your experiments are limited to the low-cost QLORA strategy. How confident are you that your linear proxy predictor, which is trained on *frozen* embeddings, would remain accurate for predicting the performance of *full* fine-tuning, a process that fundamentally alters those very embeddings?
4. Could you please discuss the relationship between COSMOS and recent, directly relevant works on efficient model selection and performance prediction, such as "Selecting large language model to fine-tune via rectified scaling law" (Li et al., 2024), "Mordal: Automated Pretrained Model Selection for Vision Language Models" (Deng et al., 2024), and "LENSLLM: Unveiling Fine-Tuning Dynamics for LLM Selection" (Shao et al., 2024)? These papers seem to address a very similar problem, and their omission is a notable gap.

---

> ### Author Response · Authors · 2025-11-28
> **Response to Reviewer MzTr (1/5)**
>
> Thank you for your time and feedback and for recognizing that the strategy selection problem we study is a practical and important challenge for researchers and practitioners! We have addressed your concerns point by point below:
>
> > W1&Q3.2: The core technical contribution of the paper feels limited in its originality. The two prediction methods instantiated within the COSMOS framework are essentially linear probing on frozen embeddings and fitting a simple exponential scaling law. These are both well-established, non-novel techniques. The contribution is thus an application of these methods rather than the development of a new prediction framework. It is unclear if the linear proxy predictor, which relies on frozen embeddings, would be at all predictive for full fine-tuning, where the embeddings themselves are updated.
>
> Thank you for raising this important point! We would like to clarify that our work does **not reuse** existing known methods, but instead proposes and instantiates **two novel predictor designs** under the COSMOS framework. To our knowledge, there are no prior methods that use the *same formulations* or *serve the same purpose* (solving the strategy selection problem via predictive modeling) as our predictors. If the reviewer has specific prior work in mind, we would greatly appreciate any pointers.
>
> Although QLoRA updates a relatively small set of parameters compared to full finetuning, in our evaluated settings it still produces substantial downstream improvements, **often 30-60% absolute accuracy gains**. This demonstrates that QLoRA is sufficiently expressive, and the changes induced by QLoRA finetuning are large enough that predicting performance gains is a meaningful and non-trivial task.
>
> While one of our QLoRA instantiation uses a linear layer as its final readout, the overall method is **not** equivalent to standard linear probing. Linear probing typically evaluates whether task labels are linearly recoverable from existing model representations [1]. In contrast, our instantiation is a **task-adaptive estimator of fine-tuning gains**, and includes several components beyond typical probing setups: (l.255-269):
> - Bidirectional transformation of a causal LM to obtain sequence-level contextual embeddings, which is not part of standard linear probing.
> - A task-specific projector that learns base (frozen) performance as a proxy signal for estimating fine-tuning performance.
> - An affine calibration step that explicitly maps proxy predictions to actual QLoRA outcomes before selection.
>
> These elements collectively form a novel performance-prediction pipeline, not a reuse of probing for representation evaluation.
>
> In Appendix O, we have also included results for **LoRA** and dense retriever:
>
> | Task | Cost Level | Pred. Acc | Act. Acc | MAE$\downarrow$ | Act. Total Cost ($) | Ours Cost ($)|  CRR $\uparrow$ (%) |
> |------|------------|----------|---------|----------|------------|-----------------|-------|
> | FPB | Low | 83.85% | 84.96% | 1.11% | 4.05  | 0.31 |  92.33% |
> | | Med | 85.34% | 86.01% | 0.67% |  12.32 | 0.31  | 97.49% |
> | | High | 85.88% | 86.10% | 0.22% | 7.84 | 0.12 | 98.52% |
>
> The results demonstrate that COSMOS instantiations also work for LoRA and dense retrievers, continue to achieve **high prediction accuracy** (MAE of **0.67%** across all cost regimes), while drastically reducing cost—even achieving a **67.6× cost reduction** in the high-cost setting.
>
> **Regarding the concern about “frozen embeddings”**: Although QLoRA/LoRA update the underlying model parameters, our empirical results show that frozen pre-finetuning embeddings—combined with our predictor pipeline already contain enough signal to estimate fine-tuning gains. Moreover, across all evaluations we deliberately use **one-round selection and minimal data** to test whether COSMOS can make reliable decisions under extreme sparsity.
>
> Even in this challenging setting, **COSMOS achieves strong performance**. As shown in Appendix G, COSMOS outperforms eight search-, training-, and scaling-law–based baselines, achieving **near-oracle selection accuracy (99.3%–100%)** while reducing computational cost by up to **49.1×** across budgets.
>
> [1] https://arxiv.org/abs/1610.01644

---

> ### Author Response · Authors · 2025-11-28
> **Response to Reviewer MzTr (2/5)**
>
> > W2&Q3.1: The choice of experimental scenarios significantly limits the practical impact of the findings. The paper focuses on Q-LoRA and retrieval-augmented ICL, which are already low-cost adaptation strategies. The real-world "strategy selection problem" is most painful when resource-intensive options like full fine-tuning are on the table. The paper motivates this with a GPT-4o case study in Appendix K but provides no empirical validation for this high-stakes, more practical scenario.
>
> Thank you for raising this important point! We would like to clarify that our motivation is in Section 3.1 l.159-166 (not in Appendix K): the computational cost of the strategy selection problem. We seek to select from a vast model--strategy--configuration space. As noted in Example 1, even with one model, two strategies, and three configurations with 10 values each to select from, we must consider thousands of combinations. Search over this vast space quickly becomes prohibitively expensive, which motivates the research problem: can we solve the strategy selection problem cost-efficiently? And thus motivates COSMOS.
>
> As stated in Appendix D, l.820-831, we focus on QLoRA (rather than full fine-tuning) for two reasons:
> - QLoRA is now a standard approach to running models on a single GPU--and is therefore of greater interest to practitioners compared to full fine-tuning or plain LoRA. In fact, part of our motivation is that *QLoRA achieves performance on par with or very close to full fine-tuning or LoRA* [2–4].
> - Validating COSMOS involved a very large number of experiments, which would be *prohibitive (in terms of compute and storage)* if we were to use full finetuning. E.g., across 5 models, 8 benchmarks, 3 seeds, 10 data portions, 5 iterations, and 5 LoRA r values, we tuned approximately 7,100 checkpoints. If we were to store all full-finetuned model checkpoints (average 26.5GB per checkpoint), this would require approximately 187TB of storage. In contrast, QLoRA checkpoints are only ~1.2GB each, reducing the total storage required to ~8.3TB, a practical scale for systematic study. The compute cost is even more prohibitive: we expended over 10,000 GPU hours to obtain the QLoRA results. Full fine-tuning would require at least 16× more GPU memory and compute [2], totaling over 162,000 GPU hours (estimated cost: 162,500 USD on 40GB A100s at USD 1/hr), which is beyond our means.
>
> [2] https://arxiv.org/abs/2305.14314
>
> [3] https://arxiv.org/abs/2310.08659
>
> [4] https://arxiv.org/abs/2407.17029

---

> ### Author Response · Authors · 2025-11-28
> **Response to Reviewer MzTr (3/5)**
>
> > W3&Q1: The formalization of the cost analysis framework in Section 3.3 is overly complex and not well-motivated. By trying to create a unified model based on fluctuating real-world prices, the framework lacks principled stability. A more robust analysis based on standardized computational units, such as FLOPs, would be more generalizable and principled, even if it's harder to map to a direct dollar amount. Could you provide a simpler, more intuitive explanation of the cost framework? Why was this complex, price-based model chosen over a more standardized and stable one like total FLOPs? How do you account for the high volatility of the prices you use (e.g., GPU spot markets on Vast.ai) in a generalizable framework?
>
> Thank you for this thoughtful question! The core intuition behind our cost framework is to **capture all costs incurred during both prediction and actual adaptation from the very start needed to obtain the final downstream performance**. Prior work often excludes critical components—such as the cost of obtaining real observations [5], performing validation [6], or running evaluations [7] which can understate total cost and lead to unfair comparisons. Our goal is to create a **complete and consistent accounting** of all effort required by an adaptation strategy.
>
> To achieve this, we define the total cost of applying a strategy on a downstream task as the sum of (Eq.2):
> 1. Adaptation cost $c_{\text{adapt}}$: the cost of running the adaptation procedure itself.
> 2. Evaluation cost $c_{\text{eval}}$: the cost of evaluating the adapted model’s performance.
>
> Similarly, for prediction, we record the **full cost** of producing a performance–cost prediction:
> - for fine-tuning strategies, the cost of training the predictor and performing calibration;
> - for ICL, the cost of generating early points for extrapolation.
> This ensures that both predicted and actual workflows are measured **end-to-end** and on equal footing.
>
> Regarding the reviewer's question on price volatility: the framework is intentionally **unit-agnostic**. We only define the **structure** of cost decomposition, but the specific units can be instantiated using **FLOPs, wall-clock time, or monetary cost**. In the paper, we used monetary cost because it provides a direct and interpretable measure for practical deployment, but switching to FLOPs or time does not change the formulation since our framework simply requires that **all end-to-end costs** be included. Crucially, cost prediction is modeled **independently** from performance prediction, so the choice of units does not affect performance estimates so the choice of units does not affect performance estimates and therefore does not influence the final strategy selection as long as all candidates are evaluated using the same cost-calculation method.
>
> [5] https://arxiv.org/abs/2005.00870
>
> [6] https://arxiv.org/abs/2406.09334
>
> [7] https://arxiv.org/abs/2403.12031

---

> ### Author Response · Authors · 2025-11-28
> **Response to Reviewer MzTr (4/5)**
>
> > W4&Q4: The related work section on scaling laws and performance prediction is incomplete. It overlooks several recent and highly relevant papers that also focus on general-purpose performance prediction for model selection using efficient trials (e.g., [8,9,10]). A discussion of these works is necessary to properly contextualize the novelty and contribution of COSMOS. Could you please discuss the relationship between COSMOS and recent, directly relevant works on efficient model selection and performance prediction, such as "Selecting large language model to fine-tune via rectified scaling law" (Li et al., 2024), "Mordal: Automated Pretrained Model Selection for Vision Language Models" (Deng et al., 2024), and "LENSLLM: Unveiling Fine-Tuning Dynamics for LLM Selection" (Shao et al., 2024)? These papers seem to address a very similar problem, and their omission is a notable gap.
>
> These are great papers! **We have added them to the revised related work (Section 2) and uploaded a new version of our manuscript.** Below, we show how COSMOS is complementary to these work:
> - **Scope of the selection problem (model-only vs. model–strategy–configuration)**: [8][10] focus primarily on model selection for fine-tuning, using rectified scaling laws to predict test loss as a function of SFT data size. [9] studies component selection inside VLMs (vision encoder + LLM backbone). COSMOS also performs model selection, but over a **strictly larger space**: it jointly selects over **model × strategy × configuration**, including training-time strategies (e.g., QLoRA, LoRA), test-time strategies (e.g., ICL variants), and rich configuration knobs (iterations, data portion, LoRA rank, shots, budgets, etc.). In this sense, [8–10] can be viewed as tackling a subset of the COSMOS selection space (model under a fixed adaptation protocol), and **their predictors could in principle be plugged in as one particular strategy-specific instantiation inside COSMOS**.
> - **Cost-awareness as a first-class objective**: The cited works are primarily designed to predict or compare performance. COSMOS, by contrast, is explicitly cost-aware: the framework models **both performance and end-to-end cost** (including the cost of adaptation, evaluation, and prediction) and then selects according to a user-specified performance–cost preference. This makes COSMOS directly applicable when practitioners must trade off quality against compute or monetary budget, a dimension not explicitly handled in [8,9,10].
> - **What is predicted: test loss vs. direct downstream performance & cost**: [8] and [10] use rectified scaling laws to predict test loss, and [9] also relies on test loss to guide VLM selection after clustering models. One challenge that COSMOS tackles *is the fact that the test loss is often not linearly correlated with downstream task performance* [11–12]. COSMOS instead directly predicts downstream performance together with cost for each strategy–model–configuration tuple. This provides a more direct signal for users who care about task-level metrics (e.g., accuracy) and allows flexible scoring based on their own performance–cost preferences.
> - **Assumptions and data efficiency**: The scaling-law approaches in [8-10] assume a monotonic relationship in test loss as data size increases, and their estimators are fitted using multiple observations across different dataset fractions and/or training durations. COSMOS’s fine-tuning instantiation does **not** impose a monotonic relationship between data size and downstream performance, which allows it to capture more irregular behaviors observed in practice. Moreover, COSMOS is designed for an extreme low-observation regime: in our experiments, we use only one real observation for QLoRA and two observations for ICL to fit the predictors, yet still obtain near-oracle selection (Section 5, Appendix G). This makes COSMOS substantially more data- and cost-efficient, and complementary to methods that rely on richer size–loss traces.
>
> In summary, [8-10] make important progress on efficient model (and component) selection via scaling laws and test-loss prediction. COSMOS builds on a different formulation: a cost-aware predictive framework for joint model–strategy–configuration selection that directly estimates downstream performance and cost under very limited observations. We will clarify these connections and complementarities in the revised manuscript.
>
> [8] https://arxiv.org/abs/2402.02314
>
> [9] https://arxiv.org/abs/2502.00241
>
> [10] https://arxiv.org/abs/2505.03793
>
> [11] https://arxiv.org/abs/2210.14199
>
> [12] https://arxiv.org/abs/2506.13216

---

> ### Author Response · Authors · 2025-11-28
> **Response to Reviewer MzTr (5/5)**
>
> > Q2: Your key insight for ICL prediction is that its performance "typically follows an exponential saturation curve". What is the empirical or theoretical basis for this specific functional form? This is a foundational assumption for half of your paper's experimental validation, and it is presented without supporting evidence or citation.
>
> Thank you for this question! The exponential-saturation behavior is **empirically observed** in our evaluation (Appendix H.3). Across 5 models, 8 benchmarks, 5 configurations, and 3 seeds, over **600 total combinations**, we consistently find that retrieval augmented ICL exhibits diminishing marginal gains as the number of shots increases. Retrieval-augmented ICL yields substantial improvements at small shot counts, but the incremental benefit decreases rapidly as more examples are added. The exponential form provides a simple and effective fit to this empirically validated pattern.
>
> Importantly, our goal is **not** to propose a new scaling-law theory for ICL. Rather, COSMOS asks a practical question:
> **Given extremely limited budget, e.g., a tiny handful of real observations, can we still make accurate strategy selections?**
> Under this constraint, we intentionally choose a lightweight functional form that can be fitted with minimal data. If more real observations were available, more complex functional forms could certainly be incorporated; COSMOS places no restriction on the predictor beyond requiring performance and cost estimates.
>
> Finally, while our experiments focus on single-round selection, COSMOS can *seamlessly incorporate multi-round updates or be paired with any search-based approach*. The exponential predictor is simply an effective instantiation in the low-observation regime, and our results demonstrate that it already guides strong and reliable strategy selection.
>
> We hopes these address your concerns and we are happy to answer any questions you may have!

---

### Official Review · Reviewer_2Juc · 2025-10-30

**Soundness:** 2
**Presentation:** 2
**Contribution:** 2
**Rating:** 4
**Confidence:** 4

**Summary:**

Summary:
This manuscript addresses the relevant and practical challenge of selecting the optimal Large Language Model (LLM) model and adaptation strategy under budget constraints. The proposed COSMOS framework offers a structured approach via predictive modeling, demonstrating significant cost savings and high prediction accuracy in its instantiations for QLoRA and retrieval-augmented ICL. The study is strengthened by its formalization of the problem and extensive experiments across diverse tasks.

**Strengths:**

1. The paper clearly formalizes the complex 'strategy selection problem' for LLMs in a multi-dimensional setting (model, strategy, configuration, task, cost).
2. COSMOS is proposed as a unified framework for joint performance and cost prediction across adaptation types.
3. Experiments cover eight benchmarks, spanning general domains and specialized areas like finance.

**Weaknesses:**

1. Limited model and strategy diversity. While COSMOS is positioned as a framework, the core empirical validation primarily relies on two models (Gemma 2B and Llama 3 8B), lacking in-depth exploration of model scale and type. The investigation into different adaptation strategies is also very limited.
2. The QLoRA predictor requires calibration on a small validation set. While the cost is included, the sensitivity of prediction accuracy to the size and composition of this calibration set is not explored. The QLoRA proxy predictor relies on a small validation set (e.g., 10% of data or ~200 examples) for affine calibration. This introduces potential brittleness: small or biased samples may lead to inaccurate performance predictions and suboptimal strategy selection. The stability of the predictor across different tasks, model scales, and validation set sizes remains unclear. Sensitivity analyses—varying validation size, sampling strategy, and dataset distribution—are needed to support the claimed robustness and ensure reliable strategy recommendations.
3. The paper primarily evaluates predictors for QLoRA and ICL. Hybrid strategies are mentioned, but the capability of COSMOS to predict performance and cost outcomes when multiple strategies are combined (applying ICL after QLoRA fine-tuning) is not demonstrated or evaluated.

4. Limited Generalization Beyond Evaluated Strategies. COSMOS is proposed as a general framework but is only evaluated on QLoRA and retrieval-based ICL. It remains unclear how easily the approach extends to other common adaptations (e.g., LoRA, full fine-tuning, adapter modules, dense/ANN retrieval, LM-based prompting pipelines, ensembling) and whether additional predictor design is required.

5. Robustness of ICL Scaling-Law Extrapolation. The ICL predictor often fits an exponential-saturation model with only two points (1-shot and 8-shot). Noisy early measurements—due to retrieval failures, query heterogeneity, or non-monotonic performance—can produce unstable extrapolations and poor strategy choices, undermining reliability in real-world noisy environments.

6. The evaluation focuses on multiple-choice/classification-style tasks; there is no evidence COSMOS predicts performance–cost for open-ended generation, code, long-context reasoning, or tool-use.

**Questions:**

1. How straightforward is it to add a new adaptation strategy (e.g., LoRA) to COSMOS? Please provide the minimal data and compute requirements to construct a new strategy-specific predictor.

2. How sensitive are the QLoRA predictor calibrations to validation set size and distribution? Is there systematic bias when the validation set is small or has distribution shifts?

3. In Table 1, several predicted values are exactly the same as the actual values when using the MAE error metric. Could this be an issue related to the choice of precision?

4. Table captions in the manuscript are incorrectly placed below the tables (a typesetting error).

---

> ### Author Response · Authors · 2025-11-28
> **Response to Reviewer 2Juc (1/5)**
>
> Thank you for your time and feedback and for recognizing the clear formalization of the strategy selection problem, proposed method, and extensive experiments! We have addressed your concerns point by point below:
>
> > W1.1&W3.1&W4.1: On the difference between the COSMOS framework and the instantiated methods proposed under the COSMOS framework
>
> Thank you for raising these questions! Many of them connect to the distinction between the **COSMOS framework** and the **specific predictors instantiated under it**, so we provide a clarification upfront.
>
> As stated in the remark at the beginning of Section 5 (l.295-302), COSMOS, as introduced in Section 3, is an **analysis framework/abstraction for approaching the model–strategy selection problem through predictive modeling**. The framework itself makes *no assumptions* about which adaptation strategies are used or how the corresponding performance and cost predictors must be designed. It formalizes how *any* compatible predictors for performance and cost can be combined to support flexible strategy selection under user-specified tradeoffs.
>
> In Section 4, we then **propose two concrete instantiations** of COSMOS for two representative and widely used adaptation strategies, QLoRA and retrieval augmented ICL. These are intended as **examples** of how COSMOS can be instantiated, not as the framework itself. The purpose is to examine *whether efficient, extremely low-observation predictors exist for prominent training-time and test-time adaptation strategies* (l.240–241), and to evaluate their effectiveness empirically in Section 5. This demonstrates that it is indeed possible to approach the model–strategy selection problem via predictive modeling.
>
> Importantly, **we do not claim that the two predictors will generalize to all possible adaptation strategies**. Our contribution is to provide (1) a **flexible and extensible formulation** for predictive model–strategy–configuration selection, (2) **two concrete instantiations** showing that the framework can be realized effectively using extremely minimal observations in widely used adaptation scenarios. This provides a foundation for future work extending COSMOS to additional adaptation methods and strategy-specific predictors.
>
> > W1.2: Limited model and strategy diversity. While COSMOS is positioned as a framework, the core empirical validation primarily relies on two models (Gemma 2B and Llama 3 8B), lacking in-depth exploration of model scale and type.
>
> Thank you for raising this important point! We would like to clarify that the experiments in Section 5 aim to evaluate the effectiveness of COSMOS instantiations for QLoRA and ICL. The *COSMOS framework/abstraction itself is not an instantiation or proposed method* as stated in our response to W1.1&W3.1&W4.1.
> **Model Diversity**:  In Appendix L, we have substantially expanded our experimental setup to include **diverse model families (Gemma, Llama, Mistral, Qwen) and newer generation models (Gemma 2), spanning various scales (2B-9B)**. We present an overview below:
> - Model pool: 5 models including Gemma 2B, Llama 3 8B, Gemma 2 9B, Qwen 2 7B, Mistral 7B v0.3
> - Strategies: QLoRA and retrieval-augmented ICL
> - QloRA configurations: Training iterations $\in$ \{4,5,6,7,8\} and data portions $\in$ \{0.1,0.2,0.3,0.4,0.5,0.6,0.7,0.8,0.9,1.0\}
> - ICL configurations: Retrieved demonstration count $\in$ \{1,2,4,8,16\}
>
> This results in **275 model--strategy--configuration combinations**, with all results averaged over three random seeds for statistical robustness.
>
> **COSMOS instantiations demonstrate strong generalization across this expanded model and strategy space**. On HellaSwag (a general-domain benchmark), it achieved an average *MAE of 0.61%*, including 0% MAE in the low-cost setting, while *reducing total cost by up to 97.78%*. On Multifin EN (a domain-specific, high-variance task), COSMOS maintained robust performance with *MAE as low as 0.45%* and consistent cost savings.
>
> | Task | Cost Level | Pred. Acc | Act. Acc | MAE$\downarrow$ | Act. Total Cost ($) | Ours Cost ($)|  CRR $\uparrow$ (%) |
> |------|------------|----------|---------|-----|----------|-----------------|-------|
> | HellaSwag | Low | 94.87% | 94.87% | 0.00% | 27.042 | 2.354 | 91.30% |
> | | Med | 94.71% | 95.64% | 0.93% | 62.838 | 2.470 |  96.07% |
> | | High | 94.94% | 95.84% | 0.90% | 124.414 | 2.759 | 97.78% |
>
> | Task | Cost Level | Pred. Acc | Act. Acc | MAE$\downarrow$ | Act. Total Cost ($) | Ours Cost ($)|  CRR $\uparrow$ (%) |
> |------|------------|----------|---------|----------|------------|-----------------|-------|
> | Multifin EN | Low | 84.09% | 85.76% | 1.67% |  0.748 | 0.271 | 63.72% |
> | | Med | 85.91% | 86.36% | 0.45% | 1.751 | 0.265 | 84.88% |
> | | High | 86.82% | 88.18% | 1.36% | 3.399 | 0.243 | 92.85% |

---

> ### Author Response · Authors · 2025-11-28
> **Response to Reviewer 2Juc (2/5)**
>
> > W1.3&W3.2&W4.2: Can COSMOS be applied to other strategies beyond QLoRA and ICL?
>
> Thank you for this thoughful question! The goal in *this work is not to enumerate every possible strategy*, but rather to enable **systematic, cost-effective strategy selection** across a *wide and customizable search space*. Previous work focus on one specific category, either focus on training-time strategy [1], or on test-time strategy [2], we are the **first** to study how to select between training-time and test-time strategy via predictive modeling.
>
> In Appendix O, we have also evaluated whether the two instantiated methods under COSMOS can be applied to other adaptation strategies such as **LoRA** and **dense retriever** (all-MiniLM-L6-v2):
>
> | Task | Cost Level | Pred. Acc | Act. Acc | MAE$\downarrow$ | Act. Total Cost ($) | Ours Cost ($)|  CRR $\uparrow$ (%) |
> |------|------------|----------|---------|----------|------------|-----------------|-------|
> | FPB | Low | 83.85% | 84.96% | 1.11% | 4.05  | 0.31 |  92.33% |
> | | Med | 85.34% | 86.01% | 0.67% |  12.32 | 0.31  | 97.49% |
> | | High | 85.88% | 86.10% | 0.22% | 7.84 | 0.12 | 98.52% |
>
> The results demonstrate that COSMOS continues to achieve **high prediction accuracy** (MAE of **0.67%** across all cost regimes), while drastically reducing cost—even achieving a **67.6× cost reduction** in the high-cost setting.
>
> These results reinforce that the **instantiated methods under the COSMOS framework are effective in the setting of LoRA and dense retrieval**. While we do not claim that these methods generalize to all possible adaptation strategies, this ablation demonstrates that COSMOS can be successfully instantiated in a further variety of strong and modern setups.
>
> As stated in l.763-l.774, the studied search space was also motivated by practicality: *evaluating each configuration requires full training and inference*, and our current study has already consumed over **10,000 GPU hours** and tuned over 7,100 checkpoints, we will open-source all checkpoints and raw results to facilitate future research.
>
> [1] https://arxiv.org/abs/2402.02314
>
> [2] https://arxiv.org/abs/2412.06540
>
> > W2&Q2: Sensitivity of QLoRA predictor calibration to validation size and distribution
>
> Thank you for raising this important point! In our experiments, the “small validation set” used for calibration is actually **just a single real observation** of downstream performance and cost. This is the **minimal viable supervision** for anchoring the affine calibration step, and we intentionally operate in this extreme low-observation regime to test whether COSMOS can still make effective selections with minimal data.
>
> Because calibration uses only one observation, there is **no variability introduced by sampling strategy or validation-set composition**—the predictor is anchored directly to the true performance of QLoRA at that specific configuration. If additional budget were available, more observations could certainly be incorporated to reduce MAE further, but this would also increase cost; COSMOS explicitly leaves this accuracy–cost trade-off to the user.
>
> Importantly, even under this minimal-data setting, COSMOS performs strongly. As shown in Appendix G, COSMOS outperforms eight search-, training-, and scaling-law–based baselines, achieving **near-oracle selection accuracy (99.3%–100%)** while reducing computational cost by **up to 49.1×** across budgets.

---

> ### Author Response · Authors · 2025-11-28
> **Response to Reviewer 2Juc (3/5)**
>
> > W5: Robustness of ICL Scaling-Law Extrapolation. The ICL predictor often fits an exponential-saturation model with only two points (1-shot and 8-shot). Noisy early measurements—due to retrieval failures, query heterogeneity, or non-monotonic performance—can produce unstable extrapolations and poor strategy choices, undermining reliability in real-world noisy environments.
>
> Thank you for raising this thoughtful point! We clarify that the exponential-saturation form used for ICL is **not arbitrary**. Across 5 models, 8 benchmarks, 5 configurations, and 3 seeds (over 600 combinations)(examples in Appendix H.3), retrieval-augmented ICL consistently shows **diminishing returns**: large gains at small shot counts, followed by smaller improvements as shots increase. The exponential-saturation curve provides a stable fit to this dominant empirical trend.
>
> It is important to note that this kind of structural assumption is standard in **all** scaling-law–based approaches: prior work similarly chooses functional forms to capture broad empirical behavior rather than to model every noisy case [1–2]. Our goal in COSMOS is also different from classical scaling-law studies. We aim to support **strategy selection** in an **extreme low-observation regime**, e.g., only *two* real ICL observations. In this setting, simple, empirically supported forms generalize far better than complex models that would overfit.
>
> Despite using only two points, COSMOS instantiations achieve **near-oracle strategy selection** and up to **49.1×** cost reduction (Appendix G). The predictor needs only to preserve the correct *relative ranking* across strategies, not to perfectly model every fluctuation within a single ICL curve.
>
> Finally, the framework is **flexible**: if users have more observations, they can adopt richer functional forms or iterative refinement. COSMOS does not fix the predictor; it only requires some mapping from observations to performance–cost predictions.
>
> We will clarify these connections to standard scaling-law assumptions in the revised manuscript.

---

> ### Author Response · Authors · 2025-11-28
> **Response to Reviewer 2Juc (4/5)**
>
> > W6: The evaluation focuses on multiple-choice/classification-style tasks; there is no evidence COSMOS predicts performance–cost for open-ended generation, code, long-context reasoning, or tool-use.
>
> Thank you for raising this important point! As noted in our response to W1.1& W3.1&W4.1, COSMOS is a general analysis framework/abstraction for the model–strategy selection problem via predictive modeling. It does *not* assume a specific task type or impose restrictions on the underlying adaptation strategy. Any method capable of predicting **performance** and **cost** can be integrated into this abstraction, whether the task is multiple-choice, open-ended generation, long-context reasoning, or others.
>
> To directly address your concern, we extend our evaluation to **open-ended mathematical reasoning (AIME2024)** using a **larger model** (s1.1-32B, a fine-tuned Qwen2.5-32B-Instruct) and **two new adaptation strategies**:
> * **s1**: force the model to continue reasoning by injecting “Wait” until the generation-budget limit [3],
> * **Self-consistency**: generate multiple reasoning traces and aggregate them under a total budget [4].
>
> We proposed and instantiate COSMOS with lightweight **scaling-law based predictors**, using **minimal real observations**:
>
> * For **s1**: $\text{Acc} = a \cdot \log(\text{budget}+1) + b$, fit with *two* points (budget = 1k, 8k).
> * For **self-consistency**: $\text{Acc} = \big[a_0 \log(k+1)+a_1\big] \cdot \log(\text{budget}+1) +
>   \big[b_0 \log(k+1)+b_1\big],$
>   fit with *four* points (k ∈ {2,5}, budgets ∈ {1k, 8k}).
>
> We evaluate a broad strategy space:
> - s1 budgets: {1k, 2k, 4k, 8k, 16k, 30k, 32k}
> - Self-consistency sample counts: {1,2,...,20,30,50,60,100,150,200,300} and per sample budget: {1k, 2k, 4k, 8k, 16k, 30k, 32k}
> - Total-token constraint per sample: ≤ 320k tokens
>
> This yields **134 strategy candidates**. Costs are measured in tokens (actual + estimated), and we use the same score function (l.374) as in the main paper. Besides single round selection as we used in our main paper, we also show how COSMOS can seamlessly combine with search-based methods and conduct multi-round selection (up to 4 rounds; stop when two consecutive rounds yield no improvement) using greedy search.
>
> COSMOS achieves **perfect strategy selection** across all 11 budget levels with **MAE = 0.0%** at every budget level and cost reduction up to 30.3×.
>
> | Task     | Budget Level (tok/sample) | Pred. Acc | Act. Acc | MAE$\downarrow$ | Act. Total Cost (k tok) | Ours Cost (k tok) | CRR $\uparrow$ (%) |
> | -------- | ------------------ | --------- | -------- | --------------- | ----------------------- | ----------------- | ------------------ |
> | AIME2024 | 2k                 | 6.0%      | 6.0%     | 0.0%            | 150                     | 48                | 67.8%              |
> |          | 5k                 | 15.0%     | 15.0%    | 0.0%            | 750                     | 129               | 82.8%              |
> |          | 8k                 | 30.0%     | 30.0%    | 0.0%            | 2280                    | 243               | 89.4%              |
> |          | 10k                | 30.0%     | 30.0%    | 0.0%            | 3150                    | 291               | 90.8%              |
> |          | 16k                | 37.7%     | 37.7%    | 0.0%            | 8640                    | 501               | 94.2%              |
> |          | 20k                | 37.7%     | 37.7%    | 0.0%            | 12600                   | 612               | 95.1%              |
> |          | 32k                | 40.0%     | 40.0%    | 0.0%            | 25380                   | 855               | 96.6%              |
> |          | 100k (Multi-round) | 50.0%     | 50.0%    | 0.0%            | 83406                   | 5274              | 91.3%              |
> |          | 200k (Multi-round) | 56.7%     | 56.7%    | 0.0%            | 183261                  | 12999             | 91.0%              |
> |          | 300k (Multi-round) | 56.7%     | 56.7%    | 0.0%            | 302448                  | 11112             | 96.3%              |
> |          | 320k (Multi-round) | 60.0%     | 60.0%    | 0.0%            | 327300                  | 29400             | 89.4%              |
>
> These results show that COSMOS extends seamlessly to **open-ended generation, larger models, and complex reasoning strategies**. This extension shows that COSMOS can be instantiated **flexibly and effectively beyond multiple-choice tasks and new strategies**, achieving **perfect selection accuracy** on AIME2024 while dramatically reducing cost.
>
> [3] https://arxiv.org/abs/2501.19393
>
> [4] https://arxiv.org/abs/2203.11171

---

> ### Author Response · Authors · 2025-11-28
> **Response to Reviewer 2Juc (5/5)**
>
> > Q1: How straightforward is it to add a new adaptation strategy (e.g., LoRA) to COSMOS? Please provide the minimal data and compute requirements to construct a new strategy-specific predictor.
>
> Thank you for the question! Adding LoRA to COSMOS is very straightforward. The process is identical to the QLoRA setup described in l.255-282, one simply replaces the one real QLoRA observation with the corresponding observation from LoRA. In Appendix O, we have already provided additional results for **LoRA** and **dense retriever** all-MiniLM-L6-v2 on Llama-3-8B-Instruct. Specifically, we varied:
> - Number of finetuning iterations $\in$ {4,5,6,7,8},
> - Data portion $\in$ {0.1,0.2,0.3,0.4,0.5,0.6,0.7,0.8,0.9,1.0}
> - Number of ICL shots $\in$ {1,2,4,8,16}
>
> This yields a total of 55 combinations. The results, shown below, demonstrate that COSMOS continues to achieve **high prediction accuracy** (MAE of **0.67%** across all cost regimes), while drastically reducing cost—even achieving a **67.6× cost reduction** in the high-cost setting:
>
> | Task | Cost Level | Pred. Acc | Act. Acc | MAE$\downarrow$ | Act. Total Cost ($) | Ours Cost ($)|  CRR $\uparrow$ (%) |
> |------|------------|----------|---------|----------|------------|-----------------|-------|
> | FPB | Low | 83.85% | 84.96% | 1.11% | 4.05  | 0.31 |  92.33% |
> | | Med | 85.34% | 86.01% | 0.67% |  12.32 | 0.31  | 97.49% |
> | | High | 85.88% | 86.10% | 0.22% | 7.84 | 0.12 | 98.52% |
>
> For a new strategy-specific predictor, providing a universal recipe for designing such predictors is beyond the scope of this work. Our aim is to take the first step and show that it is possible to directly predict downstream performance and cost, and therefore use these predictions to guide strategy selection efficiently. As stated in Section 3, COSMOS only requires *some* predictor for performance and *some* predictor for cost, and these are then plugged into the score function. In our response to W6, we further illustrate this flexibility by instantiating **two additional predictors for two new open-ended strategies**, demonstrating how new strategies can be incorporated into the framework.
>
> > Q3: In Table 1, several predicted values are exactly the same as the actual values when using the MAE error metric. Could this be an issue related to the choice of precision?
>
> Thank you for the question! This is **not** a precision or rounding issue. In Table 1, we report the actual accuracy of the strategy selected by the predictor. When the predicted best strategy matches the true best strategy, the predicted accuracy equals the actual accuracy. Therefore, identical values simply indicate that we selected the optimal strategy, not that the prediction was rounded or clipped.
>
> > Q4: Table captions in the manuscript are incorrectly placed below the tables (a typesetting error).
>
> Thank you! We have revised and uploaded the new manuscript according to your suggestions!
>
> We hope these address your concerns, and we are happy to answer any further questions you may have!

---

### Meta-Review · Area_Chair_5TEq · 2026-01-07

**Summary:**

This paper studies the strategy selection problem for LLM deployment, proposing COSMOS, a framework that predicts downstream performance and end-to-end cost to guide selection over model × adaptation strategy × configuration. The paper instantiates COSMOS for QLoRA fine-tuning and retrieval-augmented ICL, reporting substantial cost reductions with low error in the selected strategy’s achieved performance. Reviewers agree that the problem is practically relevant and that the empirical study is large-scale and carefully executed (DXmw, 2Juc). However, several reviewers raised concerns regarding limited novelty, initially narrow evaluation, and the simplicity of the instantiated predictors (MzTr, nNYi).  Overall, after considering the reviews and the rebuttal, I recommend rejection.

**Reviewer Concerns:**

Addressed: framework vs. instantiation confusion; mischaracterization of predictors; related-work omissions; additional experimental scope beyond QLoRA and ICL.

Remaining: perceived limited methodological novelty of instantiated predictors; uncertainty about robustness under minimal observations; lack of empirical validation for more expensive strategies (e.g., full fine-tuning).

**Reviewer Scores:**

I don't think reviewers will change the score.

---

### Decision · Program_Chairs · 2026-01-26

Reject